# Retinoic acid-induced protein 14 controls dendritic spine dynamics associated with depressive-like behaviors

**Soo Jeong Kim, Youngsik Woo, Hyun Jin Kim, Bon Seong Goo, Truong Thi My Nhung, Seol-Ae Lee, Bo Kyoung Suh, Dong Jin Mun, Joung-Hun Kim, Sang Ki Park\***

Department of Life Sciences, Pohang University of Science and Technology, Pohang, Republic of Korea

**Abstract** Dendritic spines are the central postsynaptic machinery that determines synaptic function. The F-actin within dendritic spines regulates their dynamic formation and elimination. Rai14 is an F-actin-regulating protein with a membrane-shaping function. Here, we identified the roles of Rai14 for the regulation of dendritic spine dynamics associated with stress-induced depressive-like behaviors. Rai14-deficient neurons exhibit reduced dendritic spine density in the *Rai14*[+/-] mouse brain, resulting in impaired functional synaptic activity. Rai14 was protected from degradation by complex formation with Tara, and accumulated in the dendritic spine neck, thereby enhancing spine maintenance. Concurrently, Rai14 deficiency in mice altered gene expression profile relevant to depressive conditions and increased depressive-like behaviors. Moreover, Rai14 expression was reduced in the prefrontal cortex of the mouse stress model, which was blocked by antidepressant treatment. Thus, we propose that Rai14-dependent regulation of dendritic spines may underlie the plastic changes of neuronal connections relevant to depressive-like behaviors.

**\*For correspondence:** skpark@postech.ac.kr

**Competing interest:** The authors declare that no competing interests exist.

## Editor's evaluation

In this manuscript, the authors discovered a new function of Rai14, an F-actin binding protein, in dendritic spine dynamics. They showed that Rai14 is localized at the spine neck and regulates spine density and function. Heterozygous Rai14 knockout mice showed impaired learning and memory and depressive-like behavior. Overall, this study provides novel insights into the molecular mechanisms underlying spine dynamics and depressive-like behavior.

## Introduction

Dendritic spines, the actin-rich protrusions on dendrites, are the major postsynaptic machinery that determines synaptic function. Owing to their unique structure consisting of a large spine head and a thin neck, dendritic spines serve as postsynaptic compartments that are biochemically and electrically separated from the dendritic shaft, thereby contributing to efficient synaptic transmission and plasticity (*Tonnesen et al., 2014*; *Yuste et al., 2000*). An imbalance between spine formation and elimination, which can result in altered spine density, can lead to synaptic hyperconnection or hypoconnection (*Forrest et al., 2018*). Importantly, an aberrant loss of dendritic spine density is closely related to diverse neuropsychiatric diseases, including major depressive disorders, schizophrenia, and neurodegenerative diseases such as Alzheimer's disease (*Forrest et al., 2018*; *Penzes et al., 2011*; *Runge et al., 2020*).

The stability of the dendritic spine is key to maintaining an appropriate number of dendritic spines. Most spines are formed during early postnatal development and undergo experience-dependent pruning during postnatal development, in which the remaining spines persist throughout life. Together with pre-existing stable spines from early development, experience-derived new stable spines provide a structural basis for life-long memory storage (*Runge et al., 2020*; *Yang et al., 2009*). When spine elimination is abnormally accelerated by environmental factors such as chronic stress and inflammation, the net spine density consequently declines along with the reduction of synapse-related genes and expression of behavioral despair (*Cao et al., 2021*; *Duman et al., 2019*; *Runge et al., 2020*). Several factors, including effectors of actin dynamics and related scaffolding proteins, have been proposed to play important roles in dendritic spine formation and shape (*Hotulainen and Hoogenraad, 2010*). However, the detailed molecular basis of synapse stability and dendritic spine maintenance requires further exploration.

Retinoic acid-induced protein 14 (Rai14) is a filamentous actin (F-actin) regulating protein with six ankyrin repeats and coiled-coil structures (*Kutty et al., 2001*; *Peng et al., 2000*). Functional studies on Rai14 illustrated its role in conferring integrity of actin filament bundles in several tissues (*Qian et al., 2013a*; *Qian et al., 2013b*). Recently, it was reported that Rai14 has membrane-shaping capability and affects dendritic branch formation (*Wolf et al., 2019*). In combination with its actin-regulatory properties, Rai14 may be related to the development of dendritic spines. However, the role of Rai14 in spine development has not yet been clarified.

In the present study, we aimed to gain insights into the function of Rai14 in the development of dendritic spines. Rai14 is stabilized by its novel interaction partner, Tara (Trio associated repeat on actin; also known as TRIO and F-actin-binding protein isoform 1), and the stabilized Rai14 specifically accumulates at the neck of dendritic spines. There, Rai14 regulates dendritic spine maintenance, consequently determining synaptic connectivity in association with stress-induced depressive-like phenotypes.

## Results

### Rai14-depleted neurons exhibit decreased dendritic spine density

To investigate the function of Rai14 in dendritic spine development, we examined dendritic morphology in Rai14-deficient mice. Since *Rai14* homozygous knockout (*Rai14*[-/-]) mice showed perinatal lethality (*Figure 1—figure supplement 1*), *Rai14* heterozygous knockout (*Rai14*[+/-]) mice were used for the in vivo experiments and behavioral analyses. Rai14-deficient neurons in mouse cortex and hippocampus displayed a significantly lower number of dendritic spines than wild-type neurons (*Figure 1A and B*).

To analyze the structure of dendritic spines, we adopted primary neuron culture system. Primary cultured hippocampal neurons from *Rai14*[-/-] embryos showed significantly lower dendritic spine density (*Figure 1C and D*) without significant differences in spine length or spine head size (*Figure 1E*). Similarly, knockdown of Rai14 in primary cortical and hippocampal neurons (*Figure 1F and G*) and P14 mouse cortical neurons (*Figure 1H*) reduced the numbers of dendritic spines, indicating that the loss of dendritic spines upon Rai14 downregulation is cell-autonomous.

### Tara stabilizes Rai14

To get a clue how Rai14 regulates dendritic spines, we examined the Rai14 protein interactome. Protein-protein interaction databases disclosed the potential association of Rai14 with Tara, another F-actin binding protein (*Huttlin et al., 2017*; *Schweppe et al., 2018*; *Seipel et al., 2001*; *Woo et al., 2019*). Rai14 and Tara share multiple additional interaction partners (*Figure 2—figure supplement 1A*), indicating that they likely form a functional complex. Indeed, a yeast two-hybrid screening using a human fetal brain cDNA library validated this interaction (*Figure 2A*), which was also confirmed by co-immunoprecipitation (co-IP) of Rai14 and Tara from mouse brain lysates (*Figure 2B*). Notably, knockdown of Tara led to the downregulation of Rai14 protein levels (*Figure 2C*), whereas overexpression of Tara brought about the upregulation of endogenous Rai14 protein levels (*Figure 2—figure supplement 1B*).

We therefore asked whether the functional effects of Rai14 on dendritic spine density were associated with Tara. First, we measured dendritic spine density upon Rai14 and/or Tara depletion (*Figure 2D*). Tara knockdown resulted in a similar reduction in dendritic spine density to those

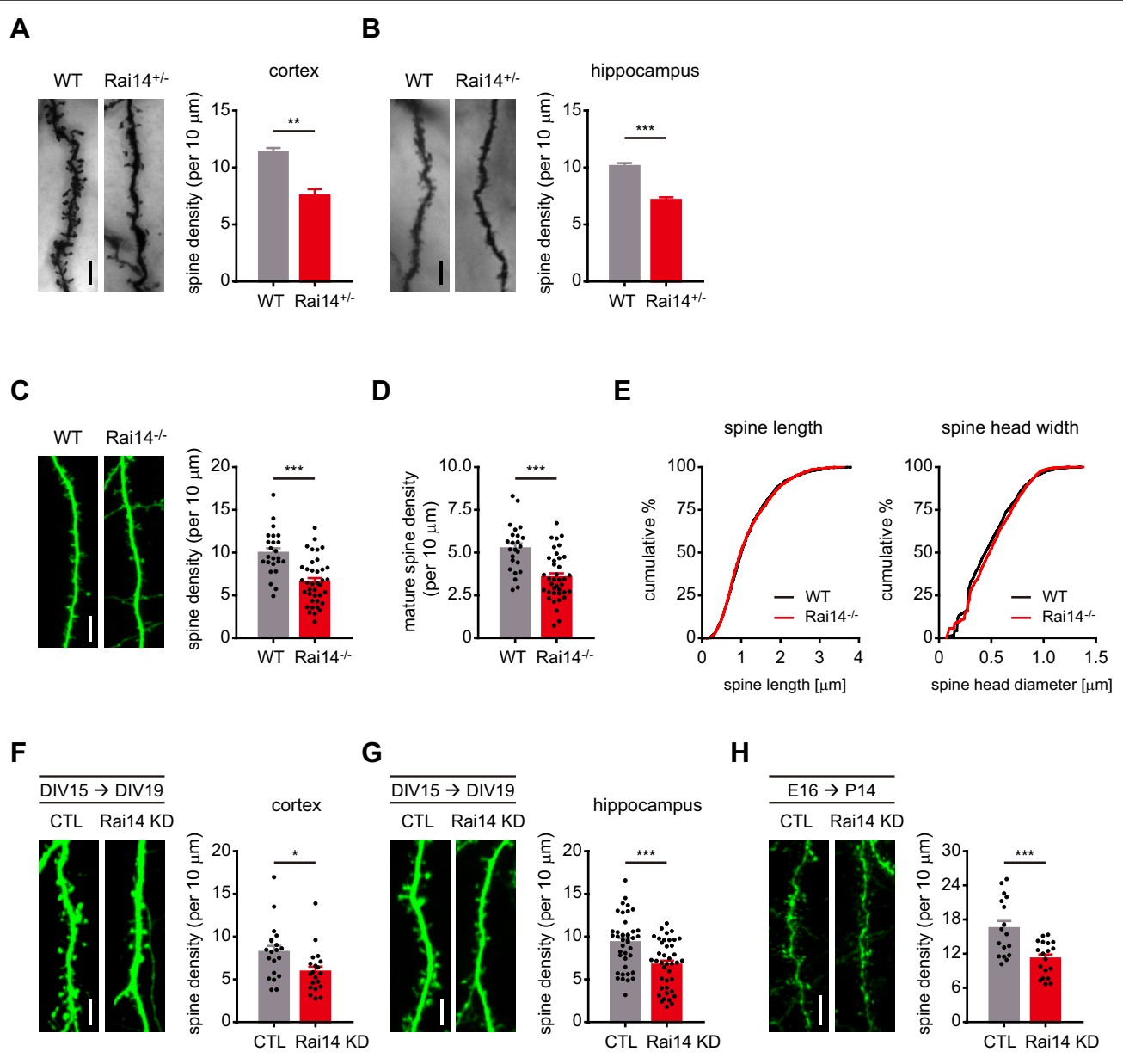

**Figure 1.** Rai14-depleted neurons exhibit decreased dendritic spine density. (**A**) Golgi-stained basal dendrites of cortical layer II/ III pyramidal neurons from adult wild-type (WT) and *Rai14+/-* mouse brains. Representative images (left) and quantitative analysis of the dendritic spine density (right) are shown (n = 4 for each group, 7–11 neurons for each mouse were analyzed). (**B**) Golgi-stained basal dendrites of hippocampal CA1 pyramidal neurons from adult WT and *Rai14+/-* mouse brains. Representative images (left) and quantitative analysis of the dendritic spine density (right) are shown. (n = 4 for each group, 8–11 neurons for each mouse were analyzed). (C–E) Dendritic spine analysis of WT and *Rai14-/-* primary cultured hippocampal pyramidal neurons (DIV19) derived from WT and *Rai14-/-* embryos. GFP-empty vector was transfected to analyze neuronal morphology. (**C**) Representative images (left) and quantitative analysis of the dendritic spine density (right) are shown. (n = 24 neurons for WT, 38 neurons for *Rai14-/-* from three separate experiments). (**D**) Quantification of mature spine density of the dendritic segments shown in (**C**). For spine type classification criteria, please see Materials and methods. (**E**) Cumulative probability plot of the spine length (left, n = 2165 spines from WT and 2207 spines from *Rai14-/-* neurons) and the maximal diameter of spine head width (right, n = 1131 mature spines from wild type, and 1210 mature spines from *Rai14-/-* neurons). (F–G) Spine density analysis of primary cultured cortical (**F**) and hippocampal (**G**) pyramidal neurons expressing scrambled shRNA (CTL) or Rai14 shRNA (Rai14 KD). Neurons were transfected at DIV15, and fixed and analyzed at DIV19–20. (**F**) Representative images of dendritic segment from cortical neurons (left) and quantitative analysis of the dendritic spine density (right) are shown (n = 20 neurons for each group from three independent cultures). (**G**) Representative images of dendritic segment from hippocampal neurons (left) and quantitative analysis of the dendritic spine density (right) are shown (n = 41 neurons for CTL, 40 neurons for Rai14 KD from 4 independent cultures). (**H**) Spine density analysis of cortical layer II/ III pyramidal neurons expressing scrambled shRNA (CTL) or Rai14 shRNA (Rai14 KD) from mouse brains. Embryos were electroporated in utero with scrambled or Rai14 shRNA at E16, and brains

*Figure 1 continued on next page*

*Figure 1 continued*

were analyzed at P14. Representative images (left) and quantitative analysis of the dendritic spine density (right) are shown (n = 17 neurons from 3 mice for CTL, 18 neurons from 3 mice for Rai14 KD). Scale bars represent 5 μm. Data are presented as mean ± SEM. *p < 0.05, **p < 0.01, and ***p < 0.001 determined by student's t-test for (**A**), (**B**), (**C**), (**D**), (**F**), (**G**) and (**H**). Kolmogorov-Smirnov test was used for (**E**). All experiments were repeated at least three times. See also *Figure 1—figure supplement 1* and *Figure 1—source data 1*.

The online version of this article includes the following source data and figure supplement(s) for figure 1:

**Source data 1.** Values for dendritic spine density analysis in Rai14-deficient groups.

**Figure supplement 1.** Loss of *Rai14* causes perinatal lethality.

**Figure supplement 1—source data 1.** Uncropped western blot images with relevant bands labeled.

expressing Rai14 shRNA. In addition, the simultaneous knockdown of Rai14 and Tara also decreased spine density to a similar extent.

To investigate how Tara regulates Rai14 protein levels, we mapped the region in Tara with Rai14 association (*Figure 2E*, *Figure 2—figure supplement 2*). The Tara$^{\Delta 241–330}$ mutant, lacking the region for Rai14 binding, failed to upregulate Rai14, whereas a fragment of Tara (aa 241–330) harboring the Rai14-binding region was sufficient to stabilize Rai14, indicating that Rai14 is stabilized by physical associations with Tara (*Figure 2F*).

We also mapped the Tara-binding region in the Rai14 protein (*Figure 2—figure supplement 3A*). The binding interface was localized to the tip of the carboxyl-terminal region containing multiple typical protein degradation motifs (*Figure 2—figure supplement 3B* and C). Indeed, a Rai14 mutant lacking the motifs in aa 948–967, Rai14$^{\Delta 948–967}$, showed significantly elevated protein levels regardless of Tara co-expression (*Figure 2G*, *Figure 2—figure supplement 3C*). As this mutant also lost capacity to interact with Tara (*Figure 2—figure supplement 3D*), it is likely that Tara stabilizes Rai14 by interfering with its degradation motifs.

Next, we tested whether Rai14 stability was directly linked to the regulation of dendritic spine density. The co-expression of Rai14 and Tara, which led to Rai14 upregulation, resulted in increased spine density, whereas the co-expression of Rai14 and Tara$^{\Delta 241–330}$ failed to increase dendritic spine density (*Figure 2H*). Moreover, unlike wild-type Rai14, the expression of Rai14$^{\Delta 948–967}$, a stabilized form of Rai14, alone was sufficient to increase spine density (*Figure 2I*), further supporting the hypothesis that Tara stabilizes Rai14 by physical interaction to positively regulate dendritic spine density.

## Tara-Rai14 complex accumulates at the neck of dendritic spines and protects spines from elimination

To understand how Tara-mediated Rai14 stabilization affects dendritic spine density, we analyzed the subcellular localization of Rai14 and Tara in neuron, especially focusing on the dendritic spines. Consistently, co-expression of Rai14 and Tara remarkably enhanced the intensity of Rai14 compared to ectopically expressed Rai14 without co-expression of Tara (*Figure 3A*, *Figure 3—figure supplement 1A*). Interestingly, when Tara was co-expressed, both Tara and Rai14 displayed a strong tendency to cluster at the neck and/or base of the dendritic spines (*Figure 3A and B*). Rai14$^{\Delta 948–967}$ also accumulated at the neck of some spines without Tara-co-expression. In contrast, when Rai14$^{\Delta ANK}$, which lacked membrane-binding ability (*Wolf et al., 2019*), was co-expressed with Tara, it did not exhibit selective accumulation at the neck of dendritic spines. The accumulation of the stabilized Rai14 at the spine neck was in a tight correlation with dendritic spine density (*Figure 3C*); Rai14$^{\Delta 948–967}$ effectively enhanced spine density whereas Rai14$^{\Delta ANK}$, which was able to interact with and was upregulated by Tara (*Figure 3—figure supplement 1B and C*), failed to increase dendritic spine density compared to Rai14 and Tara co-expressing dendrites.

Next, to examine the contribution of the Tara-Rai14 cluster for the dynamic nature of the dendritic spines, we monitored the process of spinogenesis in primary neurons co-expressing Rai14 and Tara by time-lapse imaging. Dendritic spines were grouped into Rai14-positive spines (those containing Rai14-GFP within their neck at 0 min) and Rai14-negative spines (those without Rai14-GFP at their neck) (*Figure 3D and E*). Both Rai14-positive and Rai14-negative spines underwent morphological changes, such as growth and shrinkage, with no significant temporal differences. However, the fraction of eliminated spines was markedly decreased in the Rai14-positive spines. While 33% of Rai14-negative spines were eliminated, most of the Rai14-positive spines survived, with only 2.6% of these

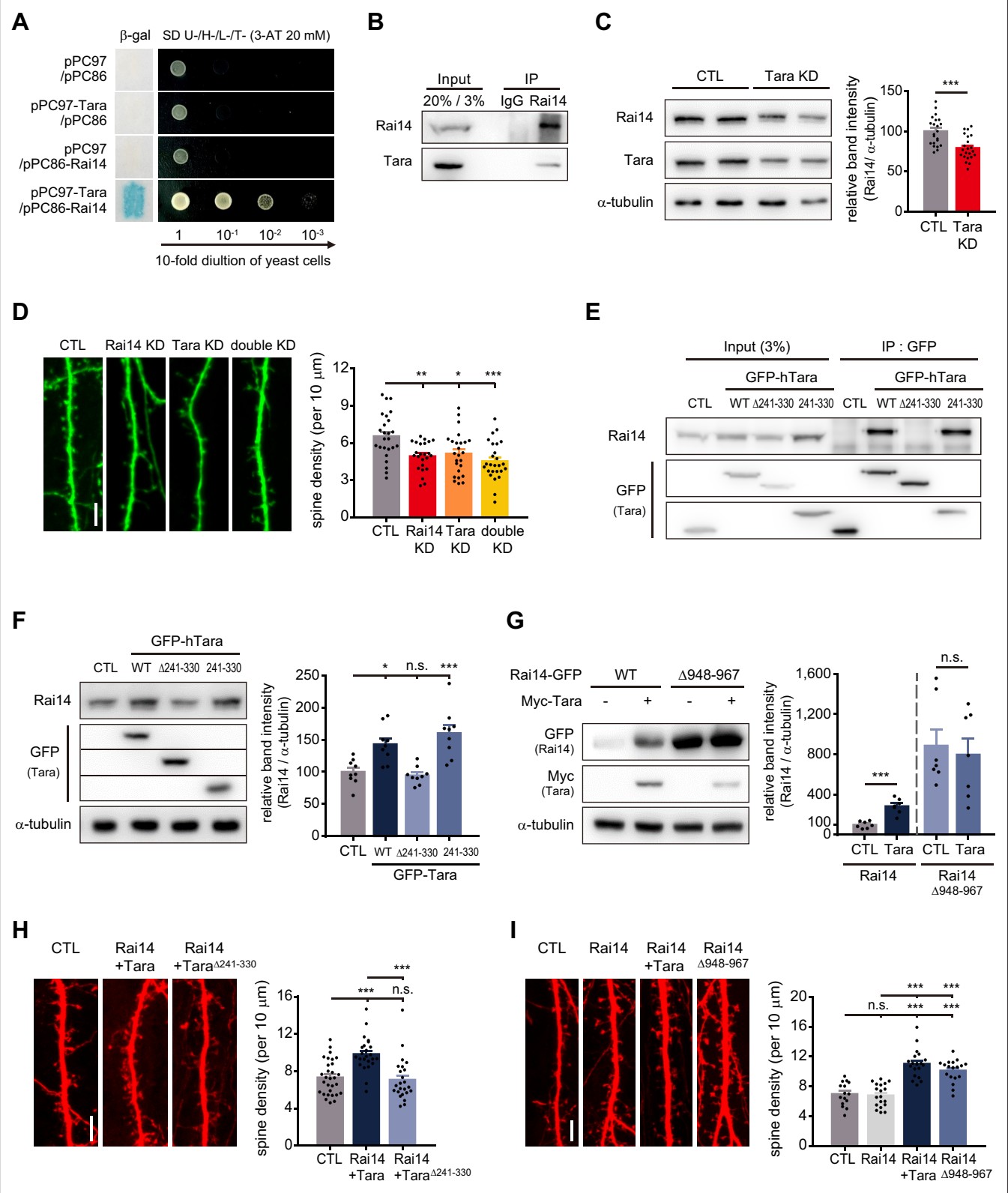

**Figure 2.** Tara-mediated stabilization of Rai14 up-regulates dendritic spine density. (**A**) Yeast two-hybrid assay of Rai14 and Tara. pPC97-Tara and pPC86-Rai14 co-transformants were analyzed byβ-galactosidase activity assay using X-gal as substrate (left) and growth on minimal media in decreasing concentrations of yeast (right). (**B**) Co-immunoprecipitation of endogenous Rai14 and Tara from P14 mouse brain lysates. (**C**) Down-regulation of Rai14 by Tara KD. Western blot image of endogenous Rai14 from HEK293 cell lysates transfected with scrambled shRNA (CTL) or Tara shRNA (Tara KD) (left)

*Figure 2 continued on next page*

*Figure 2 continued*

and relative Rai14 band intensity normalized to α-tubulin (right) are shown (n = 22 for CTL, 22 for Tara KD). (**D**) Spine density analysis of Tara and/or Rai14 KD conditions. Representative images of dendritic segments from DIV19 primary cultured hippocampal pyramidal neurons expressing indicated shRNA(s) (left) and quantification of the dendritic spine density (right) are shown (n = 25 neurons for CTL, 24 neurons for Rai14 KD, 25 neurons for Tara KD, and 26 neurons for double KD). (**E**) Localization of Tara region for interaction with Rai14. Co-immunoprecipitation of endogenous Rai14 with Tara deletion mutants was carried out in HEK293 cells. CTL: GFP-empty vector. (**F**) Up-regulation of Rai14 by Tara interaction. Western blot image of endogenous Rai14 from HEK293 cell lysates transfected with indicated plasmids (left), and relative Rai14 band intensity normalized to α-tubulin (right) are shown (n = 9). CTL: GFP-empty vector (**G**) Stabilization of Rai14 by deletion of C-terminal tip. Western blot image of Rai14 from HEK293 cell lysates transfected with indicated plasmids (left) and relative Rai14 band intensity normalized to α-tubulin (right) are shown (n = 7). (**H**) Regulation of spine density by Tara-Rai14 interaction. Representative images of dendritic segments from DIV17–19 primary cultured hippocampal pyramidal neurons expressing indicated plasmids (left) and quantification of the dendritic spine density are shown (n = 30 neurons for CTL, 24 neurons for Rai14 +Tara and Rai14 +Tara$^{\Delta241–330}$). (**I**) Regulation of spine density by Rai14 stabilization. Representative images of dendritic segments from DIV17–19 primary cultured hippocampal pyramidal neurons expressing indicated plasmids (left) and quantification of the dendritic spine density are shown (n = 15 neurons for CTL, 19 neurons for Rai14 and Rai14 +Tara, and 18 neurons for Rai14$^{\Delta948–967}$). Scale bars represent 5 μm. Data are presented as mean ± SEM. *p < 0.05, **p < 0.01, and ***p < 0.001 from student's t-test for (**C**), (**G**) and one-way ANOVA with Bonferroni's multiple comparison test for (**D**), (**F**), (**H**), and (**I**). Experiments were repeated at least three times. See also *Figure 2—figure supplements 1 and 2*, and 3, and *Figure 2—source data 1*.

The online version of this article includes the following source data and figure supplement(s) for figure 2:

**Source data 1.** Quantification on Rai14 expression and spine density in association with Tara.

**Source data 2.** Uncropped western blot images with relevant bands labeled.

**Figure supplement 1.** Rai14 and Tara form a complex.

**Figure supplement 1—source data 1.** Uncropped western blot images with relevant bands labelled.

**Figure supplement 2.** Domain mapping of Tara for interaction with Rai14.

**Figure supplement 2—source data 1.** Uncropped western blot images with relevant bands labeled.

**Figure supplement 3.** Mapping of Rai14 domain involved in Tara-mediated Rai14 upregulation.

**Figure supplement 3—source data 1.** Uncropped western blot images with relevant bands labeled.

spines eliminated. Rai14 sometimes gathered at the base and entered the neck of newly formed dendritic spines (*Figure 3F*). These spines mostly remained until the last of the imaging period, while newly formed Rai14-negative spines shrank or disappeared. To further test the role of Rai14 in spine maintenance, we induced the elimination of dendritic spines by treating neurons with latrunculin A (LatA) (*Allison et al., 1998*; *Nestor et al., 2011*; *Vlachos et al., 2009*), an actin destabilizer, and monitored spine dynamics (*Figure 3G and H*). Dendritic spines with ectopic expression of Rai14 and Tara showed significantly higher survival rates against LatA treatment than controls, indicating that Rai14 protects dendritic spines from the pressure of elimination by actin destabilization. Collectively, these data support that the Rai14-Tara complex helps in the maintenance of dendritic spines.

## Rai14 affects functional synaptic activity

Next, we attempted to see if Rai14-dependent spine maintenance contributed to excitatory synaptogenic events as dendritic spines are major postsynaptic compartments that receive most excitatory presynaptic inputs. We labeled primary hippocampal neurons with synaptophysin and PSD95, presynaptic and postsynaptic markers, respectively, to monitor excitatory synapse formation. As expected, neurons co-expressing Rai14 and Tara displayed more spines with both synaptophysin and PSD95 puncta on the spine head compared to control neurons (*Figure 4A and B*, *Figure 4—figure supplement 1A*). Within Rai14-Tara overexpressing neurons, spines containing the Rai14 cluster tended to bear synapses with higher probability than the spines without Rai14 signal (*Figure 4C*, *Figure 4—figure supplement 1B*). Concurrently, *Rai14⁻/⁻* neurons had less number of spines marked simultaneously with synaptophysin and PSD95 puncta on the spine head than wild-type neurons (*Figure 4D*). Consistently, when we evaluated the functional consequence of Rai14 depletion on synaptic transmission, miniature excitatory postsynaptic currents (mEPSCs) measured from acute brain slices of *Rai14⁺/⁻* mice showed significantly lower mean frequency without alteration of the amplitude (*Figure 4E–G*). To test further consequences of synaptic function upon Rai14 depletion, we assessed the spatial memory retention of *Rai14⁺/⁻* mice with the Morris water maze test. On the probe test day, *Rai14⁺/⁻* mice stayed significantly less in the platform-containing quadrant, whereas wild-type littermates spent more time

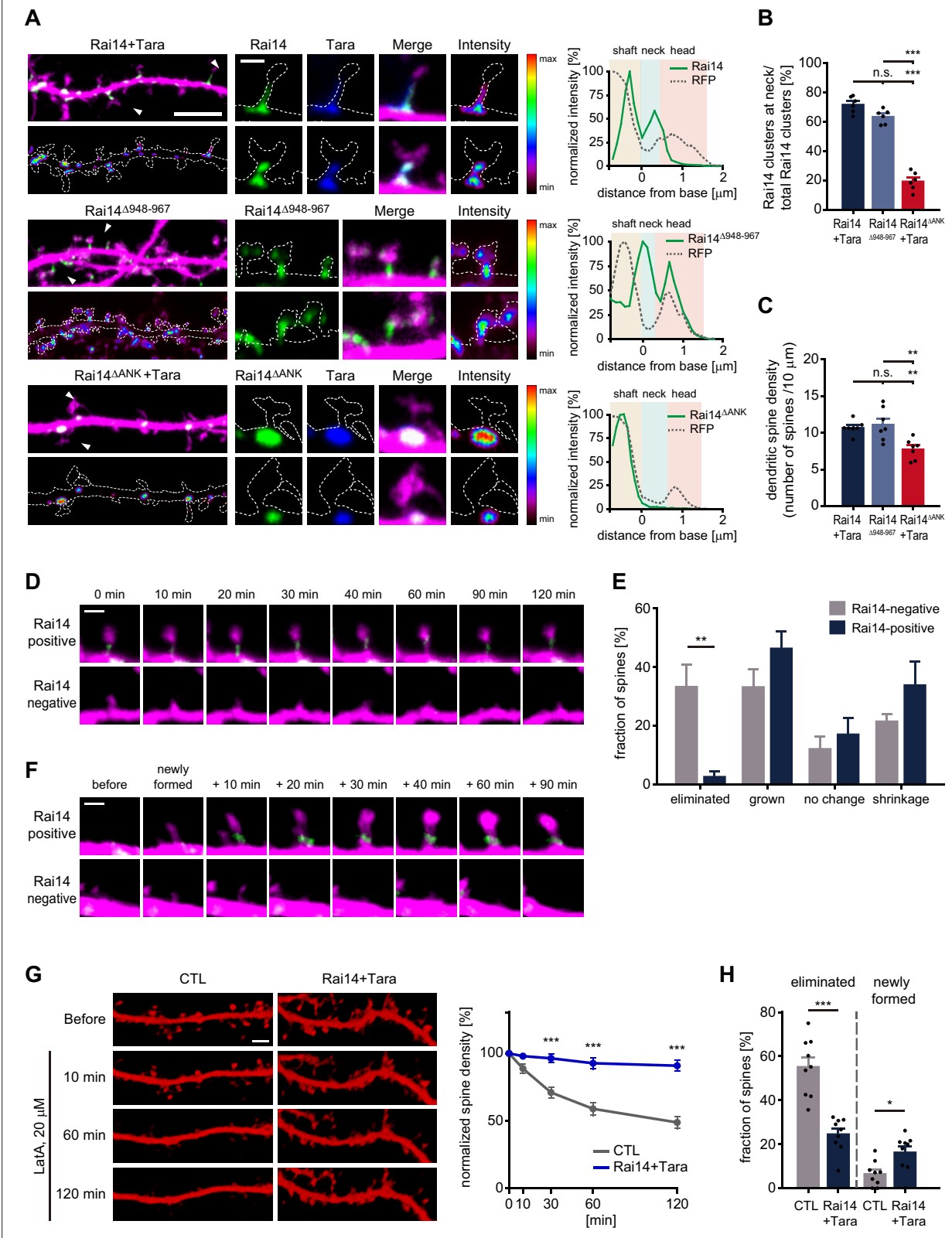

**Figure 3.** Tara-Rai14 complex accumulates at the neck of dendritic spines and protects spines from elimination. (**A–B**) Localization analyses of Rai14 at the dendritic spine. (**A**) Dendritic segments of DIV17–18 primary cultured hippocampal pyramidal neurons transfected with indicated Rai14-GFP and/or FLAG-Tara constructs are shown with an intensity heat map of Rai14 and Rai14 mutants (left, green: Rai14^WT/mut-GFP, blue: FLAG-Tara, magenta: RFP). Spines indicated by white arrowhead are shown in higher magnification with an intensity heat map of Rai14 and Rai14 mutants (middle). Representative

*Figure 3 continued on next page*

*Figure 3 continued*

intensity profiles of Rai14 and Rai14 mutants in the indicated spines are also shown (right, RFP: a morphology marker). Scale bar represents 5 μm for dendritic segments and 1 μm for magnified spine images. The contours of the dendritic shaft and spines are outlined by dashed lines. (**B**) Fraction of Rai14 clusters at spine neck relative to total Rai14 clusters within the designated dendritic segments. (n = 6 neurons) (**C**) Impact of stabilized (Rai14$^{\Delta948–967}$) or mislocalized forms of Rai14 (Rai14$^{\Delta ANK}$) expression on dendritic spine density of primary hippocampal pyramidal neurons (n = 7 neurons, DIV17–18). (**D–E**) Spine dynamics of dendritic spines with or without Rai14 from time-lapse imaging on DIV15–17 primary cultured hippocampal pyramidal neurons expressing Rai14-GFP, FLAG-Tara, and RFP. Rai14-positive spines: spines containing Rai14-GFP clusters within their neck at time 0 min, Rai14-negative spines: spines that does not contain Rai14-GFP clusters within their neck at time 0 min. (**D**) Representative images of a stable Rai14-positive spine (upper) and an eliminated Rai14-negative spine (lower). Scale bar represents 2 μm. (**E**) Quantification on the dynamics of Rai14-positive and Rai14-negative spines at 120 min compared to 0 min. (n = 5 neurons) (**F**) Representative images of newly formed dendritic spines in which Rai14-GFP recruited (upper, Rai14-positive) or not (lower, Rai14-negative) at the neck. (**G–H**) Impact of Rai14 and Tara expression on spine maintenance upon latrunculin A (LatA) treatment. (**G**) Representative images of hippocampal dendritic segments (left, morphology marker: RFP-LifeAct) and normalized spine density at indicated time points after LatA treatment (right, 20 μM) are shown (n = 9 neurons, DIV17–18). Each spine density after LatA treatment was normalized to the spine density before LatA treatment. Scale bar represents 5 μm. (**H**) Fractions of the eliminated spines and newly formed spines at 120 min time point after LatA treatment. Data are presented as mean ± SEM. *p < 0.05, **p < 0.01, and ***p < 0.001 determined by one-way ANOVA for (**B**) and (**C**), student's t-test for (**H**), and two-way ANOVA with Bonferroni's multiple comparison test for (**E**) and (**G**). All experiments were repeated at least three times. See also *Figure 3—figure supplement 1*, and *Figure 3—source data 1*.

The online version of this article includes the following source data and figure supplement(s) for figure 3:

**Source data 1.** Source data for Rai14 localization and dendritic spine dynamics.

**Figure supplement 1.** Characterization of Rai14$^{\Delta ANK}$ protein.

**Figure supplement 1—source data 1.** Uncropped western blot images with relevant bands labeled.

in the platform area (*Figure 4H–K*). In addition, *Rai14*$^{+/-}$ mice also displayed mild deficits in contextual fear memory without fear generalization (*Figure 4L–N*). On the other hand, *Rai14*$^{+/-}$ mice had no significant difference in locomotor activity and anxiety levels (*Figure 4—figure supplement 2*). These results demonstrate that the structural deficits in spine maintenance caused by Rai14 deficiency extend to functional alterations in synapses.

### *Rai14*-deficient mice exhibit depressive-like behaviors

To investigate the pathological features relevant to Rai14 depletion, we next performed RNA sequencing-based gene expression profiling on whole brains of *Rai14*-deficient mice (*Rai14*$^{+/-}$) and littermate controls followed by gene set enrichment analysis (GSEA) using curated CGP gene sets (MSigDB) (*Figure 5A and B*). Among the significant gene sets enriched in *Rai14*$^{+/-}$ mouse brains, Aston-Major Depressive Disorder_DN (the set of downregulated genes in the temporal cortex samples from patients with major depressive disorder) showed a relatively high NES rank. The distribution of the gene set was significantly enriched in the downregulated genes of *Rai14*$^{+/-}$ group, and indeed, 17 genes out of 18 significant DEGs that are included in the Aston-Major Depressive Disorder DN gene set were downregulated in *Rai14*$^{+/-}$ mouse brain (*Figure 5C and D*, *Figure 5—figure supplement 1*).

As the gene expression analysis hints at the potential link between Rai14 deficiency and depressive disorder, *Rai14*$^{+/-}$ mice were tested in depression-like behavioral paradigms. Indeed, *Rai14*$^{+/-}$ mice showed a reduced preference for sucrose solution, an anhedonic behavior (*Figure 5E*). They also exhibited longer immobile periods in the Porsolt's forced swim test, indicative of behavioral despair (*Figure 5F*), which was reversed by fluoxetine, an antidepressant, administration (*Figure 5G*). In addition, chronic fluoxetine treatment rescued loss of dendritic spines in *Rai14*$^{+/-}$ mouse brain (*Figure 5H*).

Moreover, when depressive conditions were induced to C57BL/6 mice by chronic restraint stress (CRS) (*Christoffel et al., 2011*; *Wang et al., 2017*) with validation by gained body weight (*Figure 5I*), the reductions of both Rai14 mRNA (*Figure 5J*) and protein expression (*Figure 5K*) in the prefrontal cortex were detected. However, CRS administration while receiving i.p. injections of fluoxetine failed to decrease in Rai14 expression levels. Taken together, these results support the link between the Rai14-controlled dendritic spine dynamics and depressive-like behaviors.

### Discussion

Here, we identified the function of Rai14 in the development of dendritic spines relevant to depression-like behaviors. We found that the Tara-mediated stabilization of Rai14 at the neck of developing spines

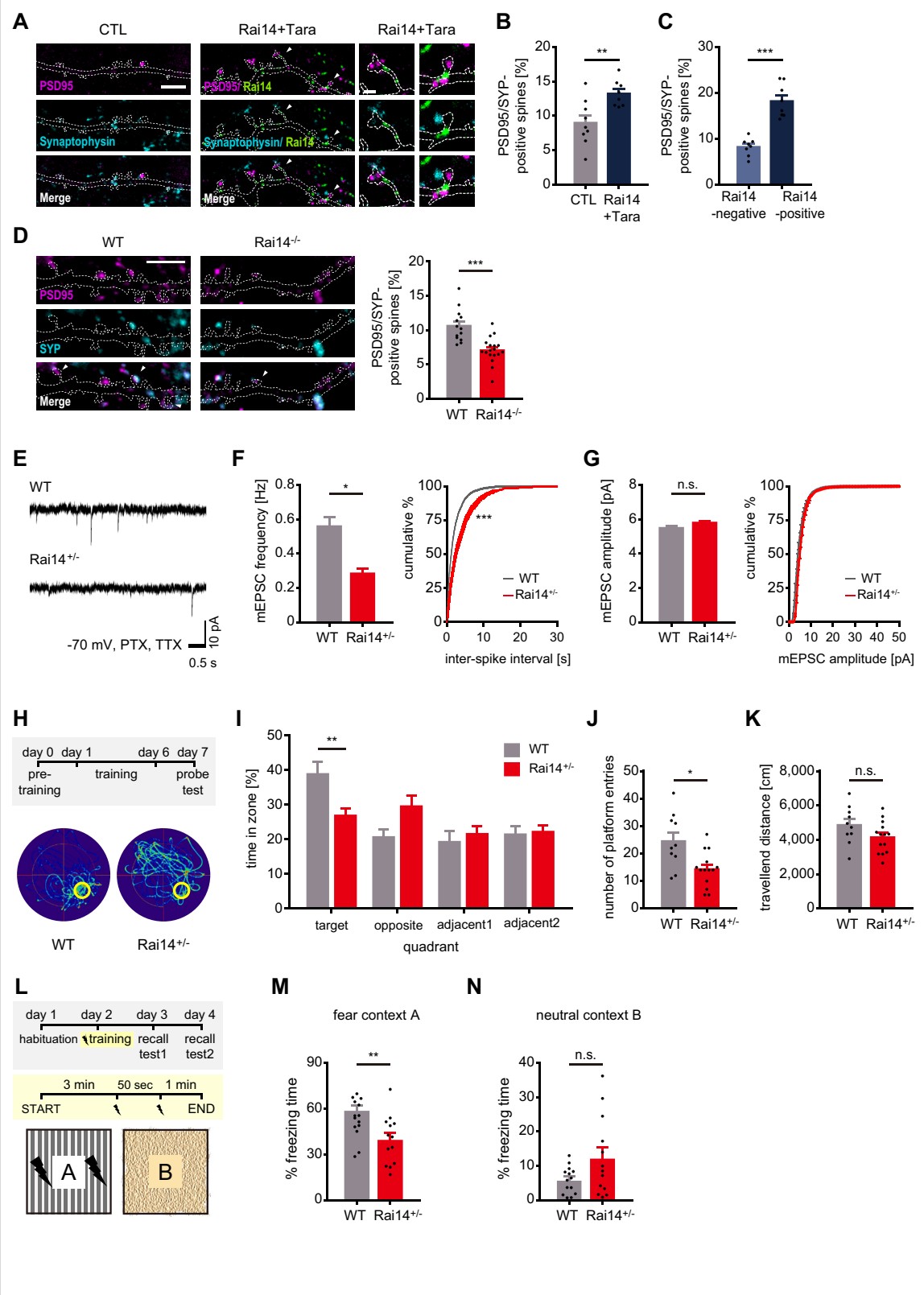

**Figure 4.** Rai14 affects functional synaptic activity. (**A–C**) Enhanced synapses in the hippocampal pyramidal neurons over-expressing Rai14 and Tara. (**A**) Representative images of DIV17–18 dendritic segments and spines are shown (magenta: PSD95, cyan: Synaptophysin, green: Rai14-GFP). Spines indicated with white arrowheads are shown in higher magnification. Scale bar represents 5 μm for dendritic segment image (left) and 1 μm for magnified spine images (right). Dashed lines indicate the contours of the dendritic shaft and spines. (**B**) Fractions of synapse-bearing spines (n = 9 neurons for CTL,

*Figure 4 continued on next page*

*Figure 4 continued*

8 neurons for Rai14 +Tara). SYP: Synaptophysin. The fraction of synaptic clusters co-localized with dendritic spines relative to entire spines was analyzed. (**C**) Fractions of synapse-bearing spines in Rai14-positive and Rai14-negative spines in hippocampal neurons expressing Rai14 and Tara (n = 8 neurons). SYP: Synaptophysin, Rai14-positive spines: spines containing Rai14-GFP clusters within their neck, Rai14-negative spines: spines without Rai14-GFP within their neck. (**D**) Decreased synapse number in the DIV18–20 hippocampal *Rai14*⁻/⁻ pyramidal neurons. The fraction of synaptic clusters co-localized with dendritic spines relative to entire spines was analyzed. Representative images of dendritic segments (left,) and fractions of synapse-bearing spines (right, n = 13 neurons for WT, 18 neurons for *Rai14*⁻/⁻). Dashed lines: contours of the dendritic shaft and spines. Scale bar: 5 µm. magenta: PSD95, cyan: SYP (synaptophysin) (**E–G**) miniature excitatory postsynaptic currents (mEPSCs) recorded from principal hippocampal CA1 pyramidal neurons of WT and *Rai14*⁺/⁻ mice. mEPSCs were recorded at –70 mV holding potential in the presence of picrotoxin (PTX) and tetrodotoxin (TTX). (**E**) Representative mEPSC traces. Scale bars represent 0.5 s and 10 pA. (**F**) Left, Average mEPSC frequency of principal hippocampal neurons from WT and *Rai14*⁺/⁻ mice. Right, Cumulative probability distributions of mEPSC inter-spike intervals (n = 3 for each group, 10–12 neurons for each mouse were analyzed). (**G**) Average (left) and cumulative probability distributions (right) of mEPSC amplitude in neurons analyzed in (**F**). (**H–K**) Morris water maze test. Performance was assessed by comparing 11- to 12-week-old male WT and Rai14⁺/⁻ mice (n = 10 for WT, 14 for *Rai14*⁺/⁻). (**H**) Experimental scheme of Morris water maze test (upper) and representative trajectories of WT and *Rai14*⁺/⁻ mice during the probe test (lower). Pre-training: training with visible platform (5 trials/ day, on day 0), training: training with hidden platform (5 trials/ day, on day 1–day 6), probe test: test with platform removed (5 min/ test, on day 7). The platform is indicated with a yellow circle. (**I**) Permanence time of WT and *Rai14*⁺/⁻ mice in indicated quadrants during the probe test. (**J**) Number of platform entries during the probe test. (**K**) Total traveled distance during the probe test. (**L–N**) Contextual fear conditioning test (n = 14 for WT, 13 for *Rai14*⁺/⁻, 11–12 week old). (**L**) Experimental scheme of the contextual fear conditioning test. In fear context A, two electric foot-shocks (0.4 mA for 1 s) were delivered with a 50 s interval. (**M**) Mean fractions of freezing time in the fear context (**N**) Mean percentage of freezing time in the neutral context. Error bars indicate the mean ± SEM. *p < 0.05, **p < 0.01, and ***p < 0.001 determined by student's t-test for (**B**), (**C**), (**D**), (**J**), (**K**), (**M**), and (**N**), two-way ANOVA with Bonferroni's multiple comparison test for (**I**). Unpaired t-test with Welch's correction was used for bar graphs, and Kolmogorov-Smirnov test was used for cumulative graphs in (**F**) and (**G**). All experiments were repeated at least three times. See also *Figure 4—figure supplement 1*, and 2, and *Figure 4—source data 1*.

The online version of this article includes the following source data and figure supplement(s) for figure 4:

**Source data 1.** Source data for synapse number and synaptic function in Rai14-deficient groups.

**Figure supplement 1.** Spine analyses by pre- and postsynaptic markers.

**Figure supplement 2.** Anxiety-related behavioral tests of *Rai14*⁺/⁻ mice.

contributed to the maintenance of mature spines. At the same time, Rai14 deficiency resulted in the loss of dendritic spines, attenuation of synaptic function, and depression-like phenotypes, including behavioral deficits relevant to mood and cognition (*Figure 6*).

## Stabilization of Rai14 by Tara

According to the data, Tara acts as a stabilizing factor for Rai14 in the Rai14-Tara complex. The Rai14 full-length protein displayed very low stability that was significantly reversed by its physical interaction with Tara. The interaction of Tara at Rai14 amino acid residues 948–967 or deletion of this region appeared to be sufficient to confer stabilization of Rai14. This type of regulatory mode is not uncommon. For example, intrinsically disordered proteins such as neuroligin 3 and tumor protein p53 are highly susceptible to 20 S proteasomal degradation; however, their specialized binding partners, S-SCAM (MAGI2) or PSD95, and MDM4, respectively, protect them from degradation (*Tsvetkov et al., 2008*; *Tsvetkov et al., 2009*). The binding of NQO1 to p53, tumor protein p73, or the ODC1 monomer also protects them from proteasomal digestion (*Tsvetkov et al., 2010*). Hsp90 shows a similar protective effect on CHEK1 (*Oh et al., 2017*). In these cases, a potential accession motif for degradation machinery resides in the interaction interface, and the interaction seems to shield the motif by interfering degradation process effectively. Indeed, favorable cleavage sites for PCSK (*Kumar et al., 2020*) and cathepsins (*Li et al., 2020*) were predicted in the region covering Rai14 amino acid residues 948–967. Moreover, large-scale protein-protein interaction studies suggested that Rai14 has interaction partners functioning in the proteasomal ubiquitination-dependent process; namely, PSMC3, UBE2V2, RBX1, LRSAM1 (*Huttlin et al., 2017*; *Schweppe et al., 2018*), and COP1 (*Szklarczyk et al., 2019*). Therefore, in addition to the regulation of Rai14 expression at the transcription level (*Fang et al., 2013*; *Gokce et al., 2009*; *Kutty et al., 2001*), a Tara-mediated post-translational regulatory mechanism for Rai14 protein expression appears to contribute to the developmental processes of the dendritic spine in the brain.

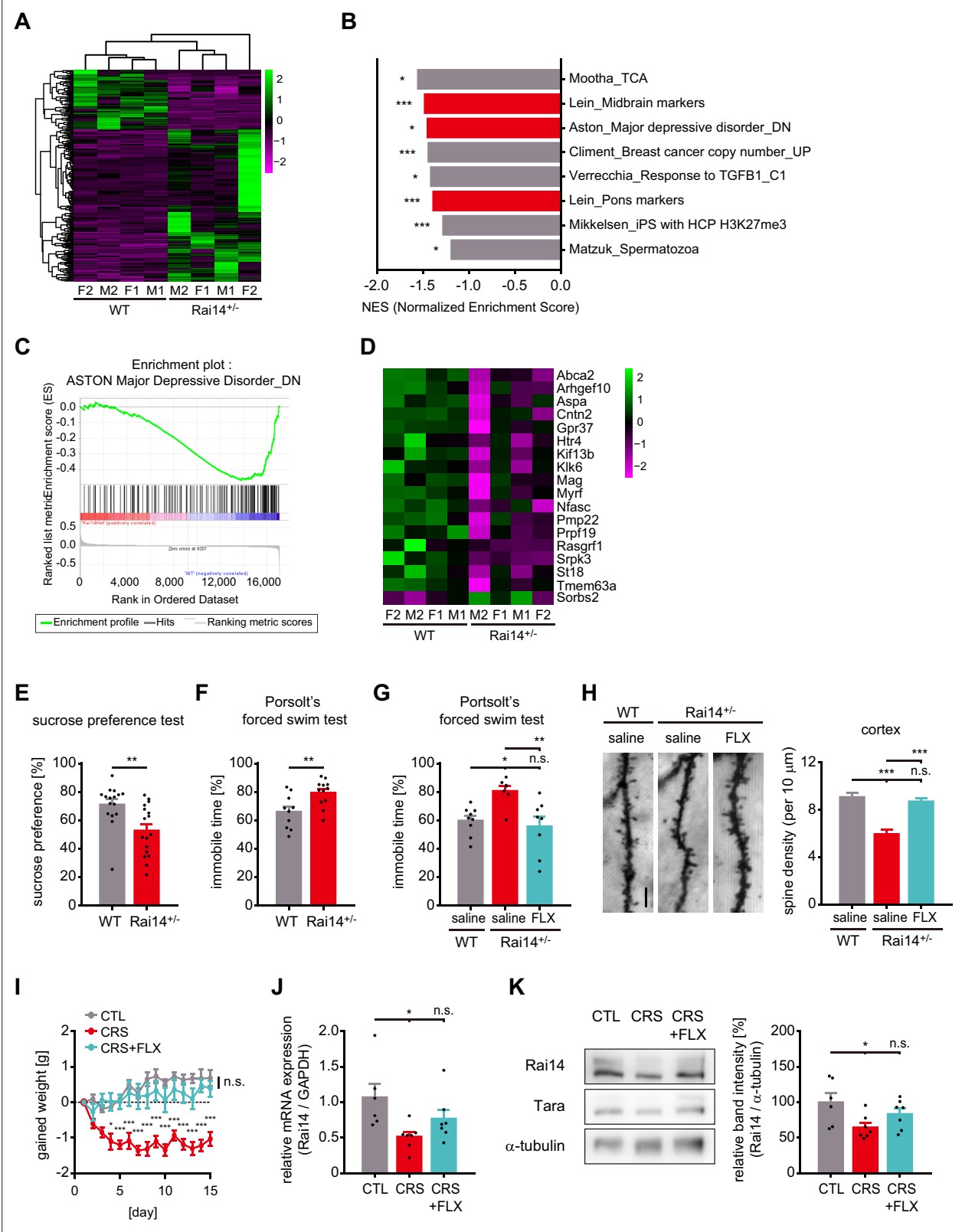

**Figure 5.** *Rai14*-deficient mice exhibit depressive-like behaviors associated with stress. (**A–D**) RNA sequencing and gene set enrichment analysis (GSEA) on whole brains of 9-week-old *Rai14+/-* and littermate controls. (**A**) Heat map of the one-way hierarchical clustering for gene expression value (log2 based normalized). 273 genes showing |fold change| ≥ 2 and raw p-value < 0.05. Green: higher expression, magenta: lower expression, F: female, M: male (n = 4 mice, 2 females + 2 males). (**B**) GSEA results using curated chemical and genetic perturbations (CGP) gene set collection from MSigDB.

*Figure 5 continued on next page*

*Figure 5 continued*

Significant gene sets (nominal p-value < 0.05) negatively enriched in *Rai14*⁺/⁻ mouse brains are listed in the order of normalized enrichment score (NES), and gene sets associated with the nervous system are indicated with red color. *p < 0.05, **p < 0.01, and ***p < 0.001. (**C**) The enrichment plot of the genes in the gene set 'Aston_Major depressive disorder_DN' generated from GSEA (*Mootha et al., 2003*; *Subramanian et al., 2005*). Upper: Profile of running enrichment score. Lower: Positions of the gene set members on the ranked ordered list. Green line: enrichment profile, black line: hits of gene set members, red zone: upregulated in *Rai14*⁺/⁻ brain, blue zone: downregulated in *Rai14*⁺/⁻ brain. (**D**) Heat map representation of transcripts included both in the 'ASTON-Major depressive disorder_DN' gene set and significant DEGs in *Rai14*⁺/⁻ mouse brains. Green: higher expression, magenta: lower expression, F: female, M: male. (**E**) Sucrose preference test. Ten- to 12-week-old male WT and *Rai14*⁺/⁻ mice were individually housed and given a free choice between 2% sucrose solution and plain water (n = 16 for WT, 17 for *Rai14*⁺/⁻). (**F**) Porsolt's forced swim test. Performance was assessed by comparing 10- to 12week old male WT and *Rai14*⁺/⁻ mice (n = 10 for WT, 12 for *Rai14*⁺/⁻). The fractions of immobile time are shown. (**G**) Porsolt's forced swim test upon anti-depressant administration. Fluoxetine (FLX, 10 mg/ kg) or saline were treated for 15 days ahead of the test (n = 9 for WT-saline, 7 for *Rai14*⁺/⁻-saline, and 8 for *Rai14*⁺/⁻-FLX) (**H**) Effects of fluoxetine (FLX) on dendritic spine density. FLX (10 mg/ kg) or saline was treated for 15 days ahead of the sampling. Representative images of Golgi-stained dendrites of cortical layer II/ III pyramidal neurons (left) and quantitative analysis of the dendritic spine density (right) are shown (n = 4 for each group, 8–12 neurons for each mouse were analyzed). (**I–K**) Effects of chronic restraint stress (CRS) and fluoxetine treatment (FLX). For CRS, C57BL/6 mice received two-hour of daily restraint stress procedures for 15 days. CRS + FLX group was administered CRS while receiving i.p. injections of 10 mg/ kg of FLX 10 min before each CRS session. (**I**) Effects of CRS and FLX on body weight gain (n = 6 for CTL, 7 for CRS, and 7 for CRS + FLX). (**J**) Relative Rai14 mRNA level in the prefrontal cortex of the mice prepared in (**I**). (**K**) Relative Rai14 protein level in the prefrontal cortex of the mice prepared in (**I**). Representative western blot image (left) and densitometric analysis of Rai14 band intensity normalized to α-tubulin (right). Error bars indicate the mean ± SEM. *p < 0.05, **p < 0.01, and ***p < 0.001 determined by student's t-test for (**E**) and (**F**), one-way ANOVA with Bonferroni's multiple comparison test for (**G**), (**H**), (**J**), and (**K**), and two-way ANOVA for (**I**). See also *Figure 5—figure supplement 1*, and *Figure 5—source data 1*.

The online version of this article includes the following source data and figure supplement(s) for figure 5:

**Source data 1.** Source data for RNA seq and depressive-like behaviors in *Rai14*⁺/⁻ mice.

**Source data 2.** Uncropped western blot images with relevant bands labelled.

**Figure supplement 1.** Alteration of gene expression profile in *Rai14*⁺/⁻ mice.

## Localization of Rai14 at the spine neck

When Tara stabilizes Rai14 in neurons, the selective accumulation of Rai14 cluster at the neck of a sub-population of dendritic spines becomes more prominent. The Rai14^ΔANK mutant loses this unique localization and just clusters at the dendritic shaft proximal to the base of dendritic spines. Because ankyrin repeats are often provided as an interface for membrane binding and protein-protein interaction (*Bennett and Baines, 2001*; *Wolf et al., 2019*), Rai14 is likely localized at the dendritic spine neck via its ankyrin repeats by binding to membrane proteins or other actin regulatory proteins within the spine neck. Furthermore, since self-assembly is one of the requirements for membrane-shaping proteins to enhance membrane curvature (*Qualmann et al., 2011*), stabilization of Rai14 by Tara co-expression or by deletion of degradation-related motifs may enhance the local Rai14 concentration required for self-assembly for larger arrays locally around the neck of the spines.

A few other proteins, including βIII-spectrin (SPTBN2) (*Efimova et al., 2017*), ankyrin-G (*Smith et al., 2014*), synaptopodin (*Deller et al., 2000*), septin 7 (*Ewers et al., 2014*), and DARPP-32 (PPP1R1B) (*Blom et al., 2013*), have been reported to localize to the spine neck. Including Rai14, these proteins share a common characteristic: actin binding and regulation of actin dynamics. Unlike the spine head filled with branched F-actin and a pool of G-actin at dynamic equilibrium (*Hotulainen and Hoogenraad, 2010*), the spine neck consists of actin in the form of a linear F-actin and periodic F-actin with a ring structure (*Bär et al., 2016*; *Bucher et al., 2020*). In particular, this periodic F-actin structure is so stable that it can give mechanical support to the spine neck. The mathematical calculation also supported that deviatoric curvature in the dendritic spine neck helps spine formation and maintenance with little force required (*Miermans et al., 2017*). That is, periodic F-actin along the spine neck can affect spine stabilization by constricting the spine neck, providing structural support for the spine head as a relatively biochemically and electrically separate compartment from the dendritic shaft (*Tonnesen et al., 2014*; *Yuste et al., 2000*). For example, ankyrin-G acts as a diffusional barrier that limits the mobility of GluA1 (GRIA1), thereby promoting AMPA receptor retention within the spine (*Smith et al., 2014*). βIII-spectrin prevents microtubule invasion into dendritic protrusions within proximal dendrites to avoid the extension of dendritic protrusions into neurites (*Fujishima et al., 2020*). In this regard, it will be of immediate interest to investigate the potential

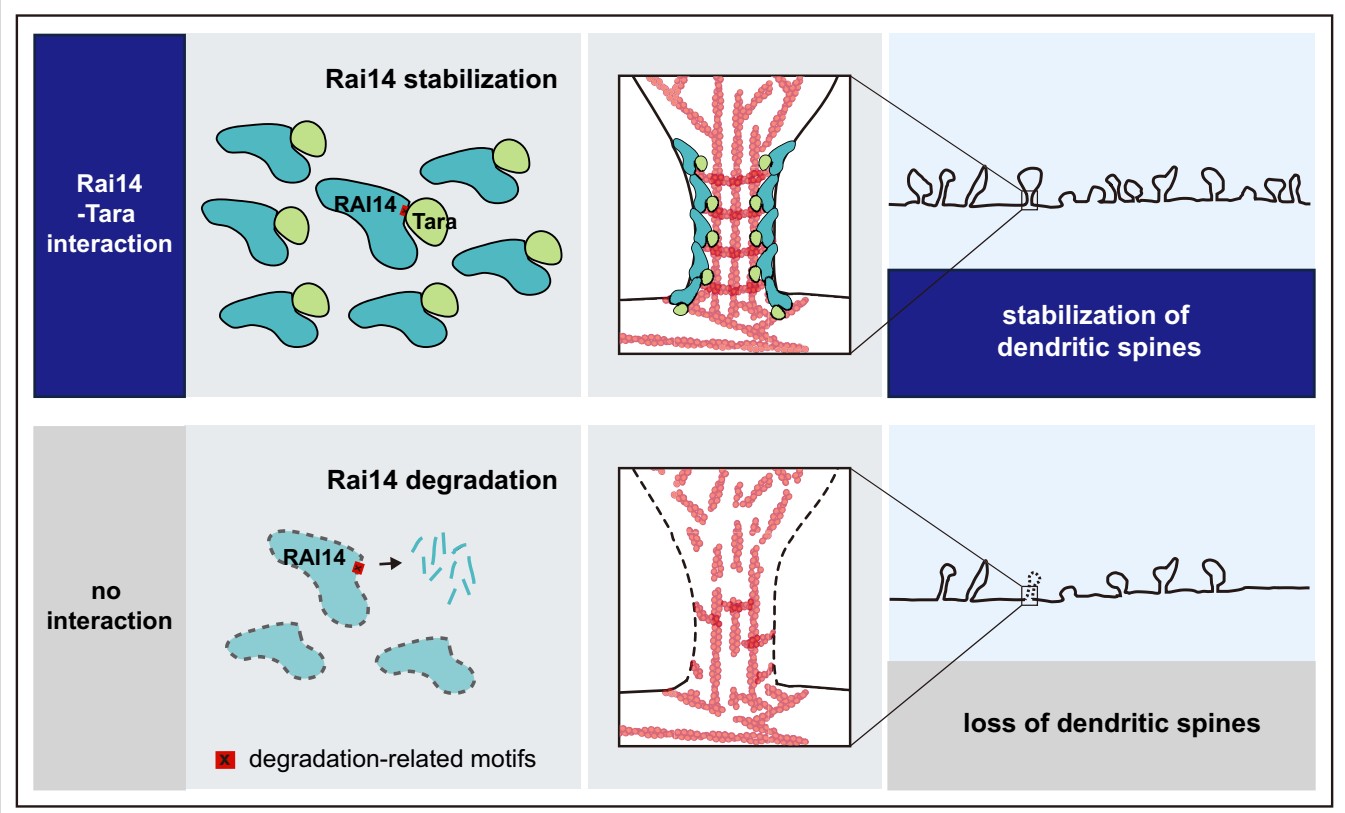

**Figure 6.** A schematic model; Tara-mediated stabilization of Rai14 for the regulation of dendritic spine dynamics Rai14-Tara interaction stabilizes Rai14 by masking degradation-related motifs within its C-terminal tip. Stabilized Rai14-Tara complex accumulates at the neck of dendritic spines via the ankyrin repeat domain of Rai14. The Rai14 cluster at the spine neck contributes to maintaining spines, thereby upregulating dendritic spine density. Rai14 deficiency leads to reduced dendritic spine density, in association with synaptic impairments relevant to depressive-like behaviors.

cooperation of Rai14 and previously known machinery for spine stabilization working at the neck to maintain the functional integrity of dendritic spines.

## Rai14 deficiency and depressive-like behaviors

Excessive, uncontrollable chronic stress is tightly linked to the expression of depressive behaviors (*Kendler et al., 1999*). In this line, chronic exposure of animals to highly stressful events is one of the well-characterized methods to establish an animal depression model that resembles clinical depression in humans (*Christoffel et al., 2011*; *Wang et al., 2017*). Therefore, it is interesting to see that a significant reduction in Rai14 expression was observed in the prefrontal cortex of mouse stress models such as the chronic restraint stress model and that Rai14-deficient mice mimicked stress-induced depressive-like behaviors, including behavioral despair, anhedonia, and cognitive deficits.

Indeed, among 18 genes that were included in both significant DEGs in Rai14[+/-] mouse brains and the Aston-Major Depressive Disorder_DN gene set, several genes were reported to be involved in dendritic spine regulation. Namely, CNTN2 (Contactin-2), GPR37 (G-protein coupled receptor 37), HTR4 (5-hydroxytryptamine receptor 4), SORBS2 (Sorbin and SH3 domain-containing protein 2; also known as ArgBP2), and RASGRF1 (Ras-specific nucleotide-releasing factor 1) are present in the dendritic spine and required for spine regulation (*Anderson et al., 2012*; *DiBattista et al., 2015*; *Lee et al., 2016a*; *Lee et al., 2016b*; *Lopes et al., 2015*; *Sonnenberg et al., 2021*; *Watkins and Orlandi, 2020*; *Zhang et al., 2016*), supporting the links between spine loss and depressive-like behaviors in Rai14 deficiency. Interestingly, bioinformatics analysis using JASPAR Predicted Transcription Factor Targets dataset predicted that half of those 18 genes, including the aforementioned spine-associated genes (CNTN2, GPR37, HTR4, SORBS2, RASGRF1, ABCA2, ASPA, MAG, PMP22) as target genes of NF-kB (*Rouillard et al., 2016*), activity-regulated transcription factor (*Kaltschmidt and Kaltschmidt, 2009*; *Snow et al., 2014*) regulated by Rai14 (*Shen et al., 2019*).

This is intriguing because previous studies have reported the potential implications of Rai14 in the BDNF and mTOR pathways, which are critical processes for depression-associated synaptic remodeling. That is, BDNF treatment increases Rai14 mRNA levels in primary striatal neurons (*Gokce et al., 2009*), and Rai14 is critical for activating mTORC1 (*Shen et al., 2019*). Conversely, studies also showed that BDNF and the mediators of mTORC1 signaling are reduced in the PFC and hippocampus after chronic stress (*Duman and Li, 2012*). Furthermore, BDNF synthesis and mTORC1 signaling activation are required for the ketamine-mediated remission of stress-induced behavioral and dendritic spine deficits (*Autry et al., 2011*; *Li et al., 2010*). In this line, the changes in BDNF expression in chronic stress conditions or upon exposure to antidepressants may alter Rai14 expression, and the altered Rai14 expression mediates the mTOR pathway and dendritic spine regulation associated with behavioral phenotypes.

Recent studies suggest that various factors, including inflammatory cytokines, neurotrophic factors, and glutamate, are associated with the neuropathology of depressive conditions (*Cao et al., 2021*; *Christoffel et al., 2011*; *Duman et al., 2019*; *Schmidt et al., 2011*). One common thing that encompasses them is that they all regulate the excitatory synapse structure. In major depressive disorder, synaptic deficits are evident, including lower numbers of synapses, synaptic connections, and reduced levels of synaptic signaling proteins (*Duman et al., 2019*). This phenomenon is well linked to the observation that a decrease in dendritic spine density in the prefrontal cortex and hippocampus is often observed in the stress-induced depression animal model, where the loss of synapses in circuits underlying affective and cognitive processes is thought to cause depressive-like behaviors (*Christoffel et al., 2011*; *Duman and Duman, 2015*). Furthermore, the degree of stress-induced spine loss in CA3 pyramidal neurons correlates significantly with memory defects in mice (*Chen et al., 2013*; *Qiao et al., 2016*). In this line, it is noteworthy that Rai14-deficient mice display a mild cognitive deficit and lower spine density, in that major depressive disorder is often associated with cognitive problems along with lower spine and synaptic densities (*Perini et al., 2019*).

## Materials and methods

### Key resources table

| Reagent type (species) or resource | Designation | Source or reference | Identifiers | Additional information |
| --- | --- | --- | --- | --- |
| Strain, strain background (*M. musculus*) | IcrTac:ICR | IMSR | Cat# TAC:icr, RRID:IMSR_TAC:icr | |
| Strain, strain background (*M. musculus*) | C57BL/6NJ | IMSR | Cat# JAX:005304 RRID:IMSR_JAX:005304 | |
| Strain, strain background (*M. musculus*) | C57BL/6NJ-Rai14em1(IMPC)J | MGI | Cat# MGI:5755416 | |
| Cell line (*H. sapiens*) | HEK293 | ATCC | Cat# PTA-4488, RRID:CVCL_0045 | |
| Antibody | Anti-Rai14 (rabbit polyclonal) | Proteintech Group | Cat# 17507–1-AP, RRID: AB_2175992 | WB (1:1,000) IP (1:1,000) |
| Antibody | Anti-Tara (rabbit polyclonal) | Thermo Fisher Scientific | Cat# PA5-29092, RRID: AB_2546568 | WB (1:1,000) |
| Antibody | Anti-PSD95, clone 7E31B8 (mouse monoclonal) | Enzo Life Sciences | Cat# ADI-VAM-PS001-E, RRID: AB_2039457 | ICC (1:50) |
| Antibody | Anti-Synaptophysin 1, Rb7.2 (rabbit monoclonal) | Synaptic Systems | Cat# 101 008, RRID: AB_2864779 | ICC (1:200) |
| Antibody | Anti-FLAG (rabbit polyclonal) | Sigma-Aldrich | Cat# F7425, RRID: AB_439687 | WB (1:2,000) ICC (1:200) |
| Antibody | Anti-FLAG, M2 (mouse monoclonal) | Sigma-Aldrich | Cat# F1804, RRID:AB_262044 | WB (1:2,000) IP (1:1,000) ICC (1:200) |
| Antibody | Anti-GFP (rabbit polyclonal) | Molecular Probes | Cat# A-11122, RRID:AB_221569 | WB (1:3,000) |
| Antibody | Anti-GFP, B-2 (mouse monoclonal) | Santa Cruz Biotechnology | Cat# sc-9996, RRID:AB_627695 | WB (1:1,000) IP (1:200) |

*Continued on next page*

*Continued*

| Reagent type (species) or resource | Designation | Source or reference | Identifiers | Additional information |
|---|---|---|---|---|
| Antibody | Anti-α-tubulin, DM1A (mouse monoclonal) | Santa Cruz Biotechnology | Cat# sc-32293, RRID:AB_628412 | WB (1:1,000) |
| Antibody | Anti-α-tubulin, 1E4C11 (mouse monoclonal) | Proteintech Group | Cat# 66031–1-Ig, RRID:AB_11042766 | WB (1:2,000) |
| Antibody | Anti-c-Myc, clone 9E10 (mouse monoclonal) | Santa Cruz Biotechnology | Cat# sc-40, RRID:AB_627268 | WB (1:1,000) |
| Antibody | Rabbit IgG, polyclonal – Isotype Control (rabbit polyclonal) | Abcam | Cat# ab37415, RRID:AB_2631996 | IP (1:1,000) |
| Antibody | Sheep Anti-Mouse IgG - Horseradish Peroxidase antibody (sheep monoclonal) | GE Healthcare | Cat# NA931, RRID:AB_772210 | WB (1:7,500) |
| Antibody | Donkey Anti-Rabbit IgG, Whole Ab ECL Antibody, HRP Conjugated | GE Healthcare | Cat# NA934, RRID:AB_772206 | WB (1:7,500) |
| Antibody | Goat anti-Rabbit IgG (H + L) cross-Adsorbed antibody, Alexa Fluor 488 (goat polyclonal) | Thermo Fisher Scientific | Cat# A-11008, RRID:AB_143165 | ICC (1:200) |
| Antibody | Goat Anti-Rabbit IgG (H + L) Antibody, Alexa Fluor 568 Conjugated (goat polyclonal) | Molecular Probes | Cat# A-11011, RRID:AB_143157 | ICC (1:200) |
| Antibody | Goat anti-rabbit IgG, Flamma 648 (goat polyclonal) | BioActs | Cat# RSA1261 | ICC (1:200) |
| Antibody | Goat anti-mouse IgG (H + L) cross-Adsobed Secondary antibody, Alexa Fluor 488 (goat polyclonal) | Molecular Probes | Cat#A-11001, RRID: AB_2534069 | ICC (1:200) |
| Antibody | Goat Anti-Mouse IgG (H + L) Antibody, Alexa Fluor 568 Conjugated (goat polyclonal) | Molecular Probes | Cat# A-11004, RRID:AB_141371 | ICC (1:200) |
| Antibody | Goat Anti-Mouse IgG (H + L) Antibody, Alexa Fluor 647 Conjugated (goat polyclonal) | Molecular Probes | Cat# A-21235, RRID:AB_141693 | ICC (1:200) |
| Recombinant DNA reagent | pEGFP-N1 | Clontech | Cat# 6085–1 | |
| Recombinant DNA reagent | pEGFP-C3 | Clontech | Cat# 6082–1 | |
| Recombinant DNA reagent | pFLAG-CMV2 | Sigma-Aldrich | Cat# E7033 | |
| Recombinant DNA reagent | pcDNA3.1/myc-His | Invitrogen | Cat# V80020 | |
| Recombinant DNA reagent | pDsRed2-N1 | Clontech | Cat# 632,406 | |
| Recombinant DNA reagent | hRai14-EGFP | This paper | N/A | Subcloned from EGFP-hRai14 |
| Recombinant DNA reagent | EGFP-hRai14 | This paper | N/A | Insertion of hRai14 CDS into pEGFP-C3 |
| Recombinant DNA reagent | FLAG-hRai14 | This paper | N/A | Insertion of hRai14 CDS into pFLAG-CMV2 |
| Recombinant DNA reagent | hRai14 $^{\Delta401\text{-}600}$-EGFP | This paper | N/A | Subcloned from hRai14-EGFP with fusion-PCR method |
| Recombinant DNA reagent | hRai14 $^{\Delta601\text{-}800}$-EGFP | This paper | N/A | Subcloned from hRai14-EGFP with fusion-PCR method |
| Recombinant DNA reagent | hRai14 $^{\Delta801\text{-}980}$-EGFP | This paper | N/A | Subcloned from hRai14-EGFP with fusion-PCR method |
| Recombinant DNA reagent | hRai14 $^{\Delta801\text{-}860}$-EGFP | This paper | N/A | Subcloned from hRai14-EGFP with fusion-PCR method |

*Continued on next page*

*Continued*

| Reagent type (species) or resource | Designation | Source or reference | Identifiers | Additional information |
|---|---|---|---|---|
| Recombinant DNA reagent | hRai14 $^{\Delta861-920}$-EGFP | This paper | N/A | Subcloned from hRai14-EGFP with fusion-PCR method |
| Recombinant DNA reagent | hRai14 $^{\Delta921-980}$-EGFP | This paper | N/A | Subcloned from hRai14-EGFP with fusion-PCR method |
| Recombinant DNA reagent | hRai14 $^{\Delta921-940}$-EGFP | This paper | N/A | Subcloned from hRai14-EGFP with fusion-PCR method |
| Recombinant DNA reagent | hRai14 $^{\Delta941-960}$-EGFP | This paper | N/A | Subcloned from hRai14-EGFP with fusion-PCR method |
| Recombinant DNA reagent | hRai14 $^{\Delta961-980}$-EGFP | This paper | N/A | Subcloned from hRai14-EGFP with fusion-PCR method |
| Recombinant DNA reagent | hRai14 $^{\Delta948-967}$-EGFP | This paper | N/A | Subcloned from hRai14-EGFP with fusion-PCR method |
| Recombinant DNA reagent | hRai14 $^{\Delta ANK}$-EGFP | This paper | N/A | Subcloned from hRai14-EGFP with fusion-PCR method |
| Recombinant DNA reagent | EGFP-hTara | *Woo et al., 2019* (PMID:31815665) | N/A | |
| Recombinant DNA reagent | FLAG-hTara | *Woo et al., 2019* (PMID:31815665) | N/A | |
| Recombinant DNA reagent | hTara-Myc | *Woo et al., 2019* (PMID:31815665) | N/A | |
| Recombinant DNA reagent | EGFP-hTara $^{1-160}$ | This paper | N/A | Subcloned from EGFP-hTara |
| Recombinant DNA reagent | EGFP-hTara $^{161-499}$ | This paper | N/A | Subcloned from EGFP-hTara |
| Recombinant DNA reagent | EGFP-hTara $^{500-593}$ | This paper | N/A | Subcloned from EGFP-hTara |
| Recombinant DNA reagent | EGFP-hTara $^{241-330}$ | This paper | N/A | Subcloned from EGFP-hTara |
| Recombinant DNA reagent | EGFP-hTara $^{\Delta161-240}$ | This paper | N/A | Subcloned from EGFP-hTara with fusion-PCR method |
| Recombinant DNA reagent | EGFP-hTara $^{\Delta241-330}$ | This paper | N/A | Subcloned from EGFP-hTara with fusion-PCR method |
| Recombinant DNA reagent | EGFP-hTara $^{\Delta331-412}$ | This paper | N/A | Subcloned from EGFP-hTara with fusion-PCR method |
| Recombinant DNA reagent | EGFP-hTara $^{\Delta413-499}$ | *Woo et al., 2019* (PMID:31815665) | N/A | |
| Recombinant DNA reagent | RFP-N1-LifeAct | *Woo et al., 2019* (PMID:31815665) | N/A | |
| Recombinant DNA reagent | EGFP-N1-LifeAct | This paper | N/A | Subcloned from RFP-N1-LifeAct |
| Recombinant DNA reagent | pLL3.7-scrambled shRNA-EGFP | *Woo et al., 2019* (PMID:31815665) | N/A | |
| Recombinant DNA reagent | pLL3.7-hTara shRNA-EGFP | *Woo et al., 2019* (PMID:31815665) | N/A | |
| Recombinant DNA reagent | pLL3.7-mTara shRNA-EGFP | This paper | N/A | Core sequence: GAAGGAGA ATGAACTCCAGTA |
| Recombinant DNA reagent | pLL3.7-hRai14 shRNA-EGFP | This paper | N/A | Core sequence: TCGGGAAA GGAATCGGTATTT |
| Recombinant DNA reagent | pLL3.7-mRai14 shRNA-EGFP | This paper | N/A | Core sequence: CGAACACT GTGGACGCCTTAA |
| Commercial assay or kit | EndoFree plasmid maxi kit | Qiagen | Cat# 12,362 | |

*Continued on next page*

*Continued*

| Reagent type (species) or resource | Designation | Source or reference | Identifiers | Additional information |
|---|---|---|---|---|
| Commercial assay or kit | FD Rapid GolgiStainTM Kit | FD Neurotechnologies | Cat# PK401 | |
| Commercial assay or kit | MAX Efficiency DH5α Competent Cells | Invitrogen | Cat# 18258012 | |
| Chemical compound, drug | Ara-C (Cytosine β-D-arabinofuranoside) | Sigma-Aldrich | C1768 | |
| Chemical compound, drug | B27 supplement | Gibco | Cat# 17504044 | |
| Chemical compound, drug | Clarity Western ECL Substrate | Bio-Rad | Cat# 1705061 | |
| Chemical compound, drug | Complete Protease Inhibitor Cocktail | Roche | Cat# 11697498001 | |
| Chemical compound, drug | DNase I | Sigma-Aldrich | Cat# DN25 | |
| Chemical compound, drug | fetal bovine serum (FBS) | Gibco | Cat# 10082147 | |
| Chemical compound, drug | Fluoxetine hydrochloride | Sigma-Aldrich | Cat# 1279804 | |
| Chemical compound, drug | Ketamine hydrochloride | Yuhan Corporation | N/A | |
| Chemical compound, drug | Laminin | Corning | Cat# 354,239 | |
| Chemical compound, drug | Latrunculin A | Cayman Chemical | Cat# CAY-10010630–2 | |
| Chemical compound, drug | Lipofectamine 2000 | Invitrogen | Cat# 11668019 | |
| Chemical compound, drug | penicillin/streptomycin | Gibco | Cat# 15140122 | |
| Chemical compound, drug | Poly-D-lysine hydrobromide | Sigma-Aldrich | Cat# P6407 | |
| Chemical compound, drug | Polyethylenimine | Polysciences | Cat# 23,966 | |
| Chemical compound, drug | RNAlaterTM Solution | Invitrogen | Cat# AM7020 | |
| Chemical compound, drug | Surgipath FSC22 Clear OCT solution | Leica Biosystems | Cat# FSC22 | |
| Chemical compound, drug | Vivamagic | Vivagen | Cat# VM001 | |
| Chemical compound, drug | Xylazine | Bayer AG | N/A | |
| Software, algorithm | ImageJ (Fiji) | *Schindelin et al., 2012* | RRID:SCR_002285 | |
| Software, algorithm | Imaris | Bitplane | RRID:SCR_007370 | |
| Software, algorithm | Olympus cellSens Software | Olympus | RRID:SCR_016238 | |
| Software, algorithm | GraphPad Prism | GraphPad | RRID:SCR_002798 | |
| Other | ECM 830 Square Wave Electroporation System | Harvard Apparatus | Cat# W3 45–0052 | Materials and methods – In utero electroporation |
| Other | Leica VT1000S vibrating blade microtome | Leica Microsystems | N/A | Materials and methods – Golgi-Cox impregnation |
| Other | Olympus Confocal Laser Scanning Microscope Fluoview FV3000 | Olympus | RRID:SCR_017015 | Materials and methods – Microscopy, Time-lapse imaging of live neurons |

## Animals

Pregnant C57BL/6 and ICR mice were purchased from Hyochang Science (Daegu, South Korea) and used for primary hippocampal neuron culture and in utero electroporation, respectively. *Rai14* knockout heterozygous mice (C57BL/6NJ-*Rai14*^em1J/J) were obtained from the Jackson Laboratory (Bar Harbor, ME). The animals were group-housed under diurnal light conditions (12 hr light, 12 hr dark cycle) and had free access to food and water. (temperature 22°C ± 2°C, humidity 50% ± 5%). Male *Rai14*$^{+/-}$ mice and wild-type littermates were kept for 10–12 weeks and subjected to behavioral analysis and brain preparation. Pregnant female *Rai14*$^{+/-}$ mice were sacrificed for primary neuron culture. All animal procedures were approved by the Institutional Animal Care and Use Committee (IACUC) of Pohang University of Science and Technology (POSTECH-2017–0037, POSTECH-2019–0025, POSTECH-2020–0008, and POSTECH-2020–0018). All experiments were carried out under the approved guidelines.

## Cell/ Neuron culture and transfection

HEK293 cells were cultured in DMEM (HyClone, South Logan, UT, USA) supplemented with 10% (v/v) fetal bovine serum (FBS) (Gibco, Gaithersburg, MD, USA) and 1% penicillin/streptomycin (Gibco) under 5% $CO_2$ at 37 °C. The cell line was authenticated using STR profiling method and tested negative for mycoplasma contamination. Cells were transfected by using either VivaMagic (Vivagen), Polyethylenimine (PEI, Polysciences, Inc, 1 mg/mL, pH 7.0), or Lipofectamine 2000 (Thermo Fisher Scientific) according to the manufacturer's instructions.

Primary cultures of hippocampal neurons were established by isolating E16–17 C57BL/6 embryonic hippocampal tissues in HBSS (Gibco) and dissociating tissues in 0.25% trypsin (Sigma-Aldrich) and 0.1% DNase I (Sigma-Aldrich) for 10 min at 37 °C. Cells were resuspended in neurobasal medium (Gibco) supplemented with 10 mM HEPES [pH7.4] and 10% (v/v) horse serum for final cell concentration being (3.0–3.5) x $10^5$ cells/mL, then plated on glass coverslips pre-coated with poly-D-lysine and laminin. Four hours after plating, the cell medium was replaced with neurobasal medium containing 2 mM glutamine, 2% (v/v) B27 supplement (Gibco), and 1% (v/v) penicillin/streptomycin. Ara-C (Sigma-Aldrich) was treated at 5 μM concentration at DIV7 for 24 h.

For primary cultures of *Rai14*$^{-/-}$ hippocampal/ cortical neurons, male *Rai14*$^{+/-}$ and female *Rai14*$^{+/-}$ mice were time-mated in order to obtain E16–E17 *Rai14*$^{-/-}$, *Rai14*$^{+/-}$, and wild-type embryos. Developing hippocampi or cortices from each embryo were separately collected, dissociated, and then plated onto pre-coated glass coverslips. Genotyping was performed by PCR using lysates from the arm, leg, and tail of each embryo.

The neurons were transfected at days in vitro (DIV) 15–17 with Lipofectamine 2000, and the medium was replaced with the culture medium 4 hr after transfection.

## Antibodies and plasmids

Anti-Rai14 rabbit polyclonal antibody (Cat# 17507–1-AP, RRID: AB_2175992) was purchased from Proteintech Group (Rosemont, IL, USA). Anti-Tara rabbit polyclonal antibody (Cat# PA5-29092, RRID: AB_2546568) was purchased from Thermo Fisher Scientific (Waltham, MA, USA). Anti-PSD95 mouse monoclonal antibody (Cat# ADI-VAM-PS001-E, RRID: AB_2039457) was purchased from Enzo Life Sciences (Farmingdale, NY, USA). Anti-Synaptophysin 1 rabbit monoclonal antibody (Cat# 101 008, RRID: AB_2864779) was purchased from Synaptic Systems (Goettingen, Germany). Anti-FLAG rabbit polyclonal and mouse monoclonal (Cat# F7425, RRID: AB_439687 and Cat# F1804, RRID: AB_262044, respectively, Sigma-Aldrich, St. Louis, MO, USA), anti-GFP rabbit polyclonal (Cat# A-11122, RRID: AB_221569, Molecular Probes, Eugene, OR, USA), anti-GFP mouse monoclonal (Cat# sc-9996, RRID: AB_627695, Santa Cruz Biotechnology, Santa Cruz, CA, USA), anti-α-tubulin mouse monoclonal (Cat# sc-32293, RRID: AB_628412, Santa Cruz Biotechnology, and Cat# 66031–1-Ig, RRID: AB_11042766, Proteintech Group), and anti-c-Myc mouse monoclonal (Cat# sc-40, RRID: AB_627268, Santa Cruz Biotechnology) were used for immunoblotting, immunoprecipitation, and immunostaining experiments. As a negative control for immunoprecipitation, normal rabbit IgG (Cat# ab37415, RRID: AB_2631996, Abcam, Cambridge, UK) was used. For immunoblotting, HRP-conjugated sheep anti-mouse IgG (Cat# NA931, RRID: AB_772210, GE Healthcare, Buckinghamshire, UK) and donkey anti-rabbit IgG (Cat# NA934, RRID: AB_772206, GE Healthcare) were used as secondary antibodies. For immunostaining, Alexa Fluor 488, Alexa Fluor 568, or Flamma 648 conjugated goat anti-rabbit IgG

(Cat# A-11008, RRID: AB_143165 and Cat# A-11011, RRID: AB_143157, Molecular Probes and Cat# RSA1261, BioActs, Incheon, South Korea) and Alexa Fluor 488 or 568 conjugated goat anti-mouse antibodies (Cat# A-11004, RRID: AB_141371 and Cat# A-21235, RRID: AB_141693, Molecular Probes) were used as secondary antibodies.

All constructs for human Rai14 were prepared by cloning hRai14 (Retinoic acid-induced protein 14) canonical isoform into pEGFP-N1, pEGFP-C3 (Clontech, Mountain View, CA, USA), and pFLAG-CMV2 (Sigma-Aldrich). To construct the deletion mutant of Rai14, regions of human Rai14 corresponding to the designated codon were amplified by PCR using Rai14-GFP plasmid as a template and cloned into pEGFP-N1 and pEGFP-C3. Constructs for human Tara were prepared by cloning full-length TRIOBP1 (Trio and F-actin binding protein1) isoform into pEGFP-C3 (Clontech, Mountain View, CA, USA), pFLAG-CMV2 (Sigma-Aldrich), and pcDNA3.1/myc-His (Invitrogen). Constructs for hTara mutants were prepared by cloning them into pEGFP-C3. Constructs for LifeAct were prepared by cloning into pEGFP-N1 and dsRed-N1. Tara and scrambled shRNA constructs were designed by cloning 19–21 nt of core sequences combined with TTCAAGAGA as the loop sequence into pLentiLox3.7 vector as described previously (*Woo et al., 2019*). Core sequences of human Tara shRNA and control scrambled shRNA were GCTGACAGATTCAAGTCTCAA and CTACCGTTGTATAGGTG, respectively. All Rai14 shRNA constructs were designed by cloning 21 nt of core sequences combined with TCTCTTGAA as the loop sequence into pLentiLox3.7 vector. The core sequence of human Rai14 shRNA and mouse Rai14 shRNA were TCG GGA AAG GAA TCG GTA TTT and CGA ACA CTG TGG ACG CCT TAA, respectively.

## Mouse lethality analysis

For lethality analysis, embryos from timed breeding of *Rai14*[+/-] mice were isolated at E17.5–E18 embryonic developmental stages and genotyped by PCR from arms, legs, and tail snips. For lethality after birth, pups from the timed mating of *Rai14*[+/-] mice were separated from their dams at P21–P28 and genotyped by PCR from tail snips.

## Golgi-Cox impregnation

Golgi-Cox impregnation was performed using an FD Rapid GolgiStain Kit (FD Neurotechnologies, Inc) according to the manufacturer's instructions. Briefly, adult *Rai14*[+/-] mice and their wild-type littermates were anesthetized with an intraperitoneal (i.p.) injection of ketamine (75 mg/kg) (Yuhan Corporation, Seoul, South Korea) and xylazine (11.65 mg/kg) (Bayer AG, Leverkusen, Germany) in PBS, and euthanized for brain isolation. Isolated brains were rinsed quickly with DW to remove blood from the surface and immersed in the impregnation solution for 14 days. Then brains were moved into solution C, and 7 days later, coronal sections in the 100 µm thickness were prepared using Leica VT1000S vibrating blade microtome (Leica Microsystems GmbH, Wetzlar, Germany). Each section was mounted on Superfrost Plus microscope slides (Fisher Scientific, Pittsburgh, PA, USA) with solution C. Excess solution was removed with pipette and filter paper. For staining, sections were rinsed with DW and then incubated with a mixture of solution D, E, and DW for 10 min. After rinsing with DW, sections were dehydrated with 50%, 75%, 95 %, and 100% ethanol. Coverslips were cleared in xylene and mounted on the section with Permount.

Images were acquired by using the FV3000 confocal laser scanning microscope (Olympus, Tokyo, Japan) and processed by using ImageJ (Fiji) software (RRID: SCR_002285, National Institute of Health, Bethesda, MD, USA).

## In utero electroporation

Pregnant ICR mice at E16 were anesthetized with an i.p. injection of ketamine (75 mg/ kg) (Yuhan Corporation, Seoul, South Korea) and xylazine (11.65 mg/kg) (Bayer AG, Leverkusen, Germany) in PBS. Rai14 shRNA or scrambled shRNA sequence in pLL3.7-EGFP vectors were purified by using EndoFree plasmid maxi kit (Qiagen, Germantown, MD, USA). Each DNA solution (2.0 µg/µL) mixed with Fast Green solution (0.001%) was injected into the lateral ventricles of the embryo through pulled microcapillary tube (Drummond Scientific, Broomall, PA, USA). Tweezer-type electrode containing two disc-type electrodes was located with appropriate angle and electric pulses were given as 35 V, 50ms, five times with 950ms intervals using an electroporator (Harvard Apparatus, Holliston, MA, USA). After

electroporation, embryos were put back into their dam's abdomen, the incision was sutured, and the mice were turned back to their home cage.

The mice were sacrificed for brain isolation at P14. Isolated brains were fixed with 4% paraformaldehyde and 10% sucrose in PBS for 24 hr, dehydrated with 10%, 20%, and 30% sucrose in PBS for more than 24 hr/session, soaked and frozen in Surgipath FSC22 Clear OCT solution (Leica Biosystems, Richmond, IL, USA). Brain tissue was sectioned by cryostats (Leica Biosystems) with 100 µm thickness, and each section was immediately bound to Superfrost Plus microscope slides (Fisher Scientific, Pittsburgh, PA, USA).

## Immunocytochemistry

Primary cultured cortical/ hippocampal neurons at DIV 17–20 were fixed with 4% paraformaldehyde and 4% sucrose in PBS for 15 min, permeabilized with 0.2% TritonX-100 in PBS for 3 min, and incubated in the blocking solution (3% BSA in PBS) for 30 min at RT. Cells were incubated with primary antibodies diluted in the blocking solution for 1.5 hr at RT or overnight at 4 °C, rinsed with PBS three times, and incubated with Alexa Fluor-conjugated secondary antibodies (Molecular Probe) diluted in the blocking solution for 1 hr at RT. Coverslips were rinsed with PBS three times and mounted in the antifade medium.

For sequential immunostaining, cells were incubated with the first primary antibody diluted in the blocking solution for 2 hr followed by two rounds of incubation with Alexa Fluor 488-conjugated secondary antibody in the blocking solution for 1 hr each at RT. Cells were rinsed with PBS for more than three times, incubated with the second primary antibody diluted in the blocking solution for 2 hr at RT, and treated with Alexa Fluor 647-conjugated secondary antibody in the blocking solution for 1 hr at RT.

## Microscopy

Dendritic spine images from primary cultured cortical or hippocampal pyramidal neurons were acquired using an FV3000 confocal laser scanning microscope (Olympus, Tokyo, Japan), with the UPLSAPO 60XO / 1.35 NA or 100 X / 1.4 NA oil-immersion objective lens and 2 x digital zoom. Images were taken in a 1024 × 1024 format, and laser power did not exceed 2% to avoid fluorescent bleaching. Stack interval of z-section was 0.56 µm for 60 x lens and 0.38 µm for 100 x lens, respectively.

Pyramidal neurons were selected by morphological guidance (*Kriegstein and Dichter, 1983*; *Luebke et al., 2010*; *Spruston, 2008*). Briefly, the structural features, such as pyramidal shape of the soma, basal dendrites originating from base of the soma, and a large main apical dendrite that descends from the apex of the soma to the tuft of dendrites, were considered in pyramidal neuron selection. Golgi-stained or transfected pyramidal neurons were randomly chosen, and dendritic spines on the nearest secondary dendritic branches from soma were analyzed (for 50–70 µm length).

In the case of Golgi-stained pyramidal neurons, TD (Transmitted light differential interference contrast) images were acquired using FV3000 confocal laser scanning microscope (Olympus, Tokyo, Japan) with the UPLSAPO 40 × 2 / 0.95 NA objective lens in a 2048 × 2048 format for dendritic spine analysis. Stack interval of z-section was 0.6 µm. Cortical layer II/ III pyramidal neurons and hippocampal pyramidal neurons were selected at motor and somatosensory cortex area, and CA1 area, respectively.

For dendritic spine images from P14 mouse brains, fluorescence images were acquired using FV3000 confocal laser scanning microscope (Olympus, Tokyo, Japan) with the UPLSAPO 60XO / 1.35 NA objective lens in a 1024 × 1024 format, with 0.43 µm z-section stack interval. Cortical pyramidal neurons were selected at layer II/ III, mostly from somatosensory cortex region.

For Rai14 localization and synapse analysis, fluorescence images were acquired using FV3000 confocal laser scanning microscope (Olympus, Tokyo, Japan), with the UPLSAPO 100XO / 1.4 NA oil-immersion objective lens in a 1024 × 1024 format, with 0.47 or 0.42 µm z-section stack interval for Rai14 localization analysis or synapse analysis, respectively.

## Image analysis

IMARIS 9.21 (Bitplane AG, Andor Technology; Belfast, Northern Ireland) was used to reconstruct z-stacks into 3D models to analyze dendritic spine images from primary cultured neurons. Dendritic spine density and spine morphology were analyzed semi-automatically using IMARIS Filament Tracer

module. Small protrusions that extended ≤4 μm from the parent dendrite were considered dendritic spines, and dendritic spines on the nearest secondary dendritic branches from soma were analyzed. For mature spine density analysis, dendritic spines were classified into three standard categories (i.e. mushroom, stubby and thin) based on the morphological characteristics of spine head width, neck width, and spine length (*Bian et al., 2015*; *Zagrebelsky et al., 2005*). Mushroom spine: maximum spine head width is greater or equal to 1.5 times of spine neck width ($D_h/D_n \geq 1.5$), stubby spine: spine head and neck are approximately of same width, and spine length is not significantly longer than the head diameter ($D_h/D_n < 2$, $L/D_h < 2$), and thin spine: maximum spine head width and spine neck width are nearly equal, and spine length is greater or equal to 2 times of maximum spine head width ($D_h/D_n < 1.5$, $L/D_h \geq 2$). Mature spines include mushroom-type spines and stubby spines. For dendritic spine analysis from Golgi-stained mouse brains, dendritic spine density was analyzed using Image J software (National Institute of Health, Bethesda, MD, USA).

For the intensity profile of Rai14-GFP within the spine, images with z-stacks were projected with maximal intensity projection, and each region of interest (ROIs) underwent line profile analysis by using cellSens software (Olympus). The set of intensity values of pixels were taken along the line with vertical stretch (from dendritic shaft 1–2 μm away from the spine to spine head) with the width of spine head width. Intensity for each channel (RFP: morphology marker, GFP: Rai14^wt/mut-GFP) was measured and individually normalized to its maximal intensity as 100% and minimal intensity as 0%.

For Rai14 distribution analysis, cellSens software (Olympus) was used to project z-stacks with maximal intensity projection. Every Rai14-GFP cluster on designated dendritic segments was counted and classified into seven classes: Rai14 at spine head +neck + base, Rai14 at spine head only, Rai14 at spine head +neck, Rai14 at spine neck only, Rai14 at spine neck +base, Rai14 at spine base only, and Rai14 at non-spine region. Non-spine Rai14 refers to the Rai14 cluster that is not connected to any spines within 0.2 μm proximity. For one neuron, 55–80 clusters were analyzed. The fraction of Rai14 at spine neck was calculated as (sum of 'Rai14 at spine head +neck + base, head +neck, neck only and neck +base' / all Rai14 clusters within the designated dendritic segment) x100.

For quantification of synapse bearing spines, images underwent deconvolution using advanced constrained iterative (CI) algorithm-based deconvolution program of cellSens software (Olympus). Co-localization of the synaptic marker with dendritic spines was determined in the merged images using ImageJ software. The fraction of synaptic clusters co-localized with dendritic spines relative to entire spines was calculated.

## Time-lapse imaging of live neurons

For live imaging on naïve state, primary hippocampal neurons were transfected with Rai14-GFP and FLAG-Tara on DIV13–15 and subjected to time-lapse imaging on DIV15–17. Live neurons were transferred to the imaging chamber (5% $CO_2$, 37 °C). Confocal images were acquired using Olympus FV3000, with the UPLSAPO 20 X / 0.75 NA objective lens in a 1024 × 1024 format with 2.5 x digital zoom. Laser power did not exceed 2% to avoid fluorescence bleaching. The stack interval of z-section was 1.04 μm. Images were taken every 10 min for 3 hr. Spines were monitored to measure fractions of spines grown, shrunk, disappeared, or with no change at 120 min. Rai14-positive spine refers to the spine containing Rai14-GFP within its neck at 0 min, whereas Rai14-negative spine refers to the spine without Rai14-GFP within its neck at 0 min.

For latrunculin A treatment, latrunculin A (LatA, CAY-10010630–2, Cayman Chemical Company) was used at a final concentration of 20 μM. Images were taken every 10 min: 20 min and 10 min before adding LatA, and 10 min – 2 hr 10 min after LatA treatment. For spine survival ratio analysis, each spine density after LatA treatment was normalized to spine density before LatA treatment. The eliminated or newly formed spine ratio was calculated as (the number of disappeared or newly formed spines at 120 min/ total spine at 0 min) x 100, respectively.

## Yeast-two hybrid screening

The human Tara coding sequence was amplified by PCR and cloned into the pPC97 vector (Invitrogen). Host *Saccharomyces cerevisiae* strain MaV203 cells were co-transformed with pPC97-Tara and human fetal brain cDNA library plasmids cloned in pPC86 (GibcoBRL). A total of $3 \times 10^6$ co-transformants was initially screened for growth on synthetic defined media (SD)-Leu⁻/ Trp⁻/ Ura⁻/ His⁻ media containing 20 mM of 3-amino-1,2,4-triazole (3-AT, Sigma-Aldrich). Plasmids were isolated from the

potential positive colonies, amplified in *Escherichia coli* DH5α, and analyzed by DNA sequencing. Colonies on SD-Leu⁻/ Trp⁻ plates were streaked onto yeast peptone dextrose (YPD) plates, and colony-lifting assays for β-galactosidase expression were carried out. For growth test on the selective media, transformants resuspended in distilled water were dropped onto a dried SD-Leu⁻/ Trp⁻/ Ura⁻/ His⁻ plate containing 20 mM 3-AT and incubated for 3 d at 30 °C.

## Western blotting and immunoprecipitation

Transfected HEK293 cells were lysed in 1 X ELB lysis buffer supplemented with 2 mM NaPPi, 10 mM NaF, 2 mM $Na_3VO_4$, 1 mM DTT, and protease inhibitor cocktail (Roche). Mouse brain tissues were isolated from anesthetized and perfused mice. Then, they were homogenized, and lysed in 1 X modified RIPA lysis buffer (50 mM Tris [pH7.5], 150 mM NaCl, 1% NP-40, 5 mM EDTA, 1% Triton X-100, 0.5% sodium deoxycholate) supplemented with 2 mM NaPPi, 10 mM NaF, 2 mM $Na_3VO_4$, 1 mM DTT, and protease inhibitor cocktail (Roche).

For western blotting, proteins were denatured by mixing lysates with 5 X SDS sample buffer (2% SDS, 60 mM Tris [pH6.8], 24% glycerol, and 0.1% bromophenol blue with 5% β-mercaptoethanol) and incubating at 95 °C for 10 min. Proteins were separated by SDS-PAGE with 8% polyacrylamide gel and transferred to PVDF membrane (Millipore, Billerica, MA, USA). Membranes were blocked with 5% skim milk in Tris-buffered saline (20 mM Tris [pH8.0], and 137.5 mM NaCl) with 0.25% Tween20 (TBST) for 30 min and incubated with primary antibodies at 4 °C for more than six hours and HRP-conjugated secondary antibodies at room temperature for more than 2 hr. Protein signals were detected by ECL solutions (BioRad, Hercules, CA, USA).

For co-IP, lysates were incubated with 1 µg of antibody at 4 °C for more than 6 hr with constant rotation. Protein-A agarose beads (Roche) washed three times with lysis buffer were mixed with IPed lysates and incubated at 4 °C for 2 hr or overnight with constant rotation. Beads were collected by centrifugation, washed three times, and mixed with SDS sample buffer for immunoblotting analysis.

## Ex vivo electrophysiology

Three-week-old mice were anesthetized with an i.p. injection of ketamine (70 mg/kg) and xylazine (5 mg/kg) in PBS, and the brains were quickly decapitated after transcardial perfusion and chilled using ice-cold carbogenated slicing solution containing 175 mM sucrose, 20 mM NaCl, 3.5 mM KCl, 1.4 mM $NaH_2PO_4$, 26 mM $NaHCO_3$, 11 mM D-(+)-glucose, and 1.3 mM $MgCl_2$ (pH 7.4). Brain slices were prepared in 350 µm thickness with a vibratome (VT1000S, Leica Microsystems GmbH, Germany) and recovered at 32 °C for 30 min in artificial cerebrospinal fluid (aCSF) (119 mM NaCl, 2.5 mM KCl, 2.5 mM $CaCl_2$, 2 mM $MgSO_4$, 1.25 mM $NaH_2PO_4$, 26 mM $NaHCO_3$, and 10 mM D-glucose while equilibrated with 95% $O_2$ and 5% $CO_2$; pH 7.3–7.4). During the recording, brain slices were placed in the recording chamber and continuously superfused with aCSF at RT. Hippocampal pyramidal neurons were selected by morphological guidance at the CA1 area. Whole-cell patch recordings in the voltage-clamp mode were controlled with a MultiClamp 700B amplifier (Molecular Devices) and acquired with a Clampex 10.7 (Molecular Devices). Recording electrodes (5–7 MΩ) were filled with a cesium-based internal solution (117 mM CsMeSO₄, 20 mM HEPES, 0.4 mM EGTA, 2.8 mM NaCl, 5 mM TEA-Cl, 2.5 mM MgATP, 0.25 mM $Na_3GTP$, and 5 mM QX-314; pH 7.2 and 275–285 mOsm adjusted with CsOH and HEPES, respectively). mEPSCs were recorded at –70 mV holding potential in the presence of 100 µM picrotoxin (Sigma) and 1 µM tetrodotoxin (Tocris). After recording, analyses were performed using Clampfit 10.7 (Molecular Devices). Briefly, spike events were manually selected to construct a template representing spike trace including several parameters, and tested build template whether clearly distinguish between noise and spikes. mEPSCs that matched the template were automatically analyzed, and the average number of events per second and peak amplitudes of events were present as frequency (Hz) and amplitude (pA), respectively.

## Mouse behavior tests

For the Morris water maze test, a large circular pool (80 cm height x 120 cm diameter) with four distinct visual cues on the wall was filled with the water (25–26 °C) to a depth of 30 cm was used as testing apparatus. Skim milk was used to make the water opaque to hide a transparent circular platform (height 28.5 cm, diameter 12 cm) submerged 1.5 cm beneath the water surface. The platform was located at a fixed position, 20 cm from the nearest pool wall throughout pre-training and training

procedures. For pre-training with visible platform and training with a hidden platform, mice were given 5 trials (maximum 1 min/ trial) per day. The entry point into the maze was changed every trial to avoid track memorization. At the end of the trial, either when the mouse had found the platform or when 60 s had elapsed, mice were allowed to rest on the platform for 40 s. One day before training with the hidden platform, mice were trained to find a visible platform with a distinct flag for habituation to the testing room and exclusion of mice with visual impairments. Following the pre-training with the visible platform, mice were trained to find the hidden platform for 6 consecutive days. In this phase, a fixed platform was hidden 1 cm below the water surface. After six-day-training with the hidden platform, mice were subjected to a probe test to evaluate memory retention. In this test, platform was removed and mice were allowed to swim for 5 min. Behavior was video recorded using a CCD camera above the pool. The time spent in each quadrant and number of platform crossings were automatically calculated by the video tracking system (SMART v2.5, Panlab).

For the contextual fear conditioning test, a cube-shaped fear conditioning chamber (26 cm x 26 cm x 24 cm) with four acrylic walls, a transparent ceiling with an empty circle in the middle, and a removable grid floor was used as testing apparatus. An infrared webcam above the chamber recorded the behavior of the mice. The chamber was within an isolation cubicle. For context A, fear conditioning chamber consisted of four opaque black walls, a transparent ceiling with a circle in the middle, and a shocker grid floor. The rods connected to a shock generating system (Panlab, Spain) delivered a current and elicited a foot-shock (0.4 mA for 1 s). For context B, which was not connected to electric foot shock, opaque black walls of the chamber was replaced with transparent acrylic walls. The shocker grid floor was removed, and a PVC floor covered with cage bedding was overlaid instead. The B chamber was scented with a peppermint odor. One day before training, mice were placed into the chamber with context A and context B for 5 min per context for habituation. During training, mice were placed in the conditioning chamber with context A for 6 min. After the first 3 min of acclimation, mice received two electric foot shocks (0.4 mA, 1 s) with a 50 s interval. On the next day, conditioned mice were placed in the same chamber for 5 min, and the freezing time was measured. Freezing was defined as the time duration of absence in all movements except for respiratory movements. For fear generalization test, fear-conditioned mice were monitored in the B chamber.

For the open-field test, the spontaneous exploratory activity of the mice in a white wooden box (60 cm x 40 cm x 20 cm) arena was assessed. Once the mouse was placed in the middle of the arena, the mouse movements in the arena was recorded for 15 min by a CCD camera above the arena and analyzed by video tracking system (SMART v2.5, Panlab).

For the elevated plus maze test, a plus-shaped maze (110 cm x 110 cm, 60 cm above the floor) with two open arms and two closed arms with 30-cm-high opaque walls was used. Once the mouse was placed in the center (5 cm x 5 cm) facing one of the closed arms, exploratory movements of the mouse in the maze were tracked and recorded for 10 min by a CCD camera above the arena and analyzed by video tracking system (SMART v2.5, Panlab).

For the sucrose preference test, mice were singly housed and subjected to the test according to previous description (*Berger et al., 2018*; *Savalli et al., 2015*; *Yu et al., 2007*). On day 1 of training, mice were deprived of food and water for 18 hr. On the day 2 and 3, food was restored, and mice were habituated to drink a 2% sucrose solution by exposure to two bottles: 2% sucrose solution and tap water for 48 hr. Then, mice were deprived of food and water for 23 hr. During the three-hour test, mice were given a free choice between two bottles: 2% sucrose solution and tap water. The position of the bottles was alternated between subjects. The weight change of the bottles before and after testing was evaluated to measure liquid intake. Sucrose preference was calculated as (sucrose solution intake / total liquid intake) x 100.

The Porsolt's forced swim test was performed according to a standard procedure. The cylindrical tank (30 cm height x 20 cm diameter) was filled with water (25°C–28 °C) up to a height of 15 cm. Sets of dividers (47 cm height x 23 cm depth) were placed between the tanks to prevent mice from seeing each other during the test. Each mouse was gently placed in the water, and activity was monitored for 6 min by video recording. The tanks were refilled with clean water after each test session. For the analysis, the time that each mice spent mobile during the last four minutes of the test was measured. Mobility was defined as any movements other than those necessary to balance the body and keep the head above the water. Immobile time was calculated as (total 240 s-mobile time). For forced swim test with *Rai14*[+/-] mice with fluoxetine treatment, mice were singly housed for 7 days before fluoxetine or

saline treatment. *Rai14*+/--Fluoxetine group received i.p. injection of 10 mg/kg fluoxetine hydrochloride (Sigma Aldrich, USP) in the volume of 10 ml/kg at every 11:30 a.m. for 16 days, whereas WT-Saline and *Rai14*+/--Saline group received i.p. injection of 0.9% saline in the volume of 10 ml/kg. Three hours after the last injection, the mice were subjected to the forced swim test.

## Chronic restraint stress (CRS) and Fluoxetine treatment

Seven-week-old male C57BL/6 mice were divided into three groups: control, CRS + Saline, CRS + Fluoxetine, and singly housed for 7 days before CRS treatment. For CRS, mice were placed into a 50 ml polypropylene conical tube (BD Falcon, 352070) with 11 holes for air ventilation. A paper towel was fixed just behind the mouse to prevent further movement in the tube. Restraint stress was introduced to mice for 2 hr per day (11:00–13:00) for 15 days. For the CRS group, mice were administered CRS while receiving i.p. injections of 10 mg/kg Fluoxetine hydrochloride (Sigma Aldrich, USP) or 0.9% saline in the volume of 10 ml/kg before each CRS session. For the control group, mice received i.p. injections of 0.9% saline and were put back to their home cage. Twenty-four hr after the last stress session, the mice were euthanized and the brains were isolated. Hippocampal and prefrontal cortical tissues from the left hemisphere were kept in RNAlater Solution (Invitrogen) and stored at –80 °C for later RNA preparation; ones from the right hemisphere were stored at –80 °C for later protein preparation.

## Quantitative real-time PCR (qRT-PCR)

The mouse brains were homogenized in TRI-Solution (Bio Science Technology) to extract total RNA according to manufacturer's instructions. Total RNA was quantified photometrically, and underwent reverse transcription with ImProm-II Reverse transcriptase (Promega Corporation). Quantitative real-time PCR (qRT-PCR) was performed using FastStart Universal SYBR Green Master (Roche) and the StepOnePlus thermocycler (Applied Biosystems). The relative expression among the groups was calculated using $2^{-\Delta\Delta Ct}$ method. The primer sequences used were mouse Rai14: forward GTGGATGT GACTGCCCAAGA/ reverse TTTCCCCGAGTTGTCAATGT, mouse GAPDH: forward CACTGAAAG GGCATCTTGG/ reverse TTACTCCTTGGAGGCCATG.

## RNA-sequencing and bioinformatics analysis

The brains from two 9-week-old male *Rai14*+/- mice and two male wild-type littermate controls and two 9-week-old female *Rai14*+/- mice and two female wild-type littermate controls were used for RNA-sequencing. The mice were deeply anesthetized with isoflurane inhalation and transcardially perfused with PBS. Then brains except the cerebellum and pons (left cerebral hemispheres) were isolated and kept in RNAlater Solution (Invitrogen) with dry ice and subjected for further RNA-Seq library construction and transcriptome sequencing.

RNA-Seq library construction, transcriptome sequencing, and expression profiling were performed by Macrogen (Macrogen, Inc, Seoul, Korea, http://www.macrogen.com/). Briefly, the mRNA from each brain sample was pooled for RNA-Seq library construction using TruSeq Stranded mRNA LT Sample Prep Kit (Illumina, San Diego, CA, USA). The mRNA library was subjected to paired-end transcriptome sequencing (Illumina platform). Raw RNA-Seq reads were trimmed with a quality cutoff Q30, and trimmed reads were mapped and aligned to the reference genome (mm10) using HISAT2. The aligned reads were then subjected to transcript assembly and quantification using StringTie program. Gene expression levels were calculated based on the read count, transcript length, and depth of coverage using FPKM (Fragments Per Kilobase of transcript per Million mapped reads), RPKM (Reads Per Kilobase of transcript per Million mapped reads), and TPM (Transcripts Per Kilobase Million) methods. Differential gene expression (DEG) analysis of *Rai14*+/- and wild-type groups was performed using DESeq2. Genes with nbnom WaldTest raw *P*-value < 0.05 were considered to be significant. The hierarchical clustering heat map was performed with "R" program (https://cran.r-project.org/) (Team RC. R: A language and environment for statistical computing. Vienna Austria: R Foundation for Statistical Computing; 2014).

GSEA (v4.1.0, Broad Institute)(*Mootha et al., 2003*; *Subramanian et al., 2005*) was performed using the entire ranked list of the expression data set determined from RNA sequencing on whole brains of wild-type and *Rai14*+/- mice. Gene sets were obtained from curated chemical and genetic perturbations (CGP) gene set collection from from MSigDB (*Liberzon et al., 2015*; *Liberzon et al.,*

*2011*; *Subramanian et al., 2005*). GSEA calculated whether genes within a gene set are randomly distributed, enriched at the top or bottom of the ranked list. Significant gene sets from curated CGP gene sets were determined using the nominal p-values. Normalized enrichment scores and p-values were measured to find enrichments with statistical significance (p < 0.05).

## Statistical analysis

All graphs were presented as the mean ± SEM. All statistical analyses were performed using GraphPad Prism 5.0 software. Statistical significance of the data was analyzed by two-tailed Student's t-test for comparisons between two groups and one-way or two-way ANOVA followed by Bonferroni's post-hoc test for comparisons among multiple groups.

# Acknowledgements

This work was supported by the funds from the National Research Foundation of Korea (NRF-2021R1A2C3010639, NRF-2020M3E5E2039894, and NRF-2017R1A5A1015366 to S.K.P.). This research was also supported by the KBRI basic research program funded by the Korean Ministry of Science and ICT (21-BR-03–01 to S.K.P.) and Basic Science Research Program funded by the Ministry of Education (2020R1A6A3A01096024 to Y.W.).

# Additional information

## Funding

| Funder | Grant reference number | Author |
|---|---|---|
| National Research Foundation of Korea | NRF-2021R1A2C3010639 | Sang Ki Park |
| National Research Foundation of Korea | NRF-2020M3E5E2039894 | Sang Ki Park |
| National Research Foundation of Korea | NRF-2017R1A5A1015366 | Sang Ki Park |
| Ministry of Science and ICT, South Korea | 21-BR-03-01 | Sang Ki Park |
| Ministry of Education | 2020R1A6A3A01096024 | Youngsik Woo |

The funders had no role in study design, data collection and interpretation, or the decision to submit the work for publication.

## Author contributions

Soo Jeong Kim, Conceptualization, Investigation, Methodology, Project administration, Resources, Validation, Visualization, Writing – original draft, Writing – review and editing; Youngsik Woo, Funding acquisition, Investigation, Resources, Validation, Visualization, Writing – review and editing; Hyun Jin Kim, Investigation; Bon Seong Goo, Investigation, Validation, Writing – review and editing; Truong Thi My Nhung, Bo Kyoung Suh, Investigation, Writing – review and editing; Seol-Ae Lee, Dong Jin Mun, Validation, Writing – review and editing; Joung-Hun Kim, Writing – review and editing; Sang Ki Park, Conceptualization, Funding acquisition, Supervision, Writing – original draft, Writing – review and editing

## Author ORCIDs

Soo Jeong Kim http://orcid.org/0000-0002-6640-9445
Youngsik Woo http://orcid.org/0000-0002-8308-8532
Hyun Jin Kim http://orcid.org/0000-0001-9108-151X
Bo Kyoung Suh http://orcid.org/0000-0001-8079-9446
Sang Ki Park http://orcid.org/0000-0003-1023-7864

## Ethics

All of the animals were handled according to approved Institutional Animal Care and Use Committee (IACUC) of Pohang University of Science and Technology (POSTECH-2017-0037, POSTECH-2019-0025,

POSTECH-2020-0008, and POSTECH-2020-0018). All experiments were carried out under the approved guidelines.

### Decision letter and Author response
Decision letter https://doi.org/10.7554/eLife.77755.sa1
Author response https://doi.org/10.7554/eLife.77755.sa2

---

## Additional files

### Supplementary files
• Transparent reporting form

• Source data 1. Unedited raw western blot images in *Figure 2* and *Figure 5*.

• Source data 2. Unedited raw western blot images in *Figure 1—figure supplement 1*, *Figure 2—figure supplements 1 and 2*.

• Source data 3. Unedited raw western blot images in *Figure 2—figure supplement 3*, and *Figure 3—figure supplement 1*.

### Data availability
Source data files including the numerical data associated with the figures are provided (for figures 1, 2, 3, 4, and 5). The source data files with original uncropped western blot images are also provided as PDF files (figures with the uncropped gels with relevant band labelled) and a zipped folder (the original files of the raw unedited gels). Sequencing data have been deposited at Dryad (doi:https://doi.org/10.5061/dryad.1rn8pk0w9).

The following dataset was generated:

| Author(s) | Year | Dataset title | Dataset URL | Database and Identifier |
|---|---|---|---|---|
| Kim S, Woo Y, Kim H, Goo B, Nhung T, Lee S, Suh B, Mun D, Kim J, Park S | 2022 | Data from: Retinoic acid-induced protein 14 controls dendritic spine dynamics associated with depressive-like behaviors | https://doi.org/10.5061/dryad.1rn8pk0w9 | Dryad Digital Repository, 10.5061/dryad.1rn8pk0w9 |

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
