## [Editor Report]

In this manuscript, the authors discovered a new function of Rai14, an F-actin binding protein, in dendritic spine dynamics. They showed that Rai14 is localized at the spine neck and regulates spine density and function. Heterozygous Rai14 knockout mice showed impaired learning and memory and depressive-like behavior. Overall, this study provides novel insights into the molecular mechanisms underlying spine dynamics and depressive-like behavior.

---

## [Decision Letter]

**Decision letter after peer review:**

Thank you for submitting your article "Retinoic acid-induced protein 14 controls dendritic spine dynamics associated with depressive-like behaviors" for consideration by *eLife*. Your article has been reviewed by 2 peer reviewers, and the evaluation has been overseen by a Reviewing Editor and Lu Chen as the Senior Editor. The following individual involved in the review of your submission has agreed to reveal their identity: Pirta Hotulainen (Reviewer #2).

Essential revisions:

It is not necessary to perform additional experiments, but please revise the manuscript as follows:

1) Please add some critical missing experimental details, as outlined in reviewer #1's comments, such as which neurons they are analyzing in Figure 1.

2) Please modify the text so that the authors do not overstate their results. Please see reviewer #1 and #2's comments.

*Reviewer #1 (Recommendations for the authors):*

1. The Figure legends are generally lacking in experimental details, including the brain regions, cell types, and mouse ages analyzed.

2. The authors should provide some explanation of how Rai14 might be regulating gene expression. The gene expression analysis was done on whole brain WT and Rai14+/- mice. Does the brain region matter?

*Reviewer #2 (Recommendations for the authors):*

In my opinion the experiments and results very well support the conclusions and the results are well presented. If I would add here something it could be Rai14 localization in dendritic neck imaged with super-resolution techniques, possibly together with actin. This would be interesting but not necessary. It would (maybe) give ideas of mechanistic details, such as, is Rai14 localizing periodically in spine necks like actin rings.

This sentence can be rephrased:

Lines 236 -239: "share a common characteristic: actin binding and regulation of actin dynamics. Unlike the spine head filled with branched F-actin and a pool of G-actin at dynamic equilibrium (Hotulainen and Hoogenraad, 2010), the spine neck consists of actin in the form of a linear F-actin bundle and periodic F-actin with a ring structure (Bar et al., 2016; Bucher et al., 2020). "

I agree that F-actin is mostly linear or in rings in necks but I would be cautious to say linear F-actin bundle. Thus, I would drop the word "bundle" out from this sentence.

---

## [Author Response]

Essential revisions:It is not necessary to perform additional experiments, but please revise the manuscript as follows:1) Please add some critical missing experimental details, as outlined in reviewer #1's comments, such as which neurons they are analyzing in Figure 1.

We added experimental details in both figure legends and the Materials and methods section.

In figure legends, we added information on;

– brain regions and neuron type analyzed (Figure 1A, 1B, and 1H, Figure 4E–G, Figure 5H).

– neuron type and analysis time points (Figure 1C–E, 1F–1G, Figure 2D, 2H, and 2I, Figure 3A, 3C 3D–E, and 3G, Figure 4A–C and 4D).

– mouse age analyzed (Figure 4H–K, and 4L–N, Figure 5A–D, 5E, and 5F).

– the amount of fluoxetine injected (Figure 5G, 5H, and 5I–K).

In Materials and methods;

–We elaborated on the details of Microscopy and Image analysis.

2) Please modify the text so that the authors do not overstate their results. Please see reviewer #1 and #2's comments.

We modified texts to avoid potential overstatements.

– Abstract, line 18: “Rai14-deficient neurons failed to maintain a proper dendritic spine density in the Rai14+/- mouse brain,” – “Rai14-deficient neurons exhibit reduced dendritic spine density in the Rai14+/- mouse brain,”

– Result, line 64: “Rai14-depleted neurons fail to maintain a normal number of dendritic spines” – “Rai14-depleted neurons exhibit decreased dendritic spine density”

– Figure 1 title: “Rai14-depleted neurons fail to maintain a normal number of dendritic spines” – “Rai14-depleted neurons exhibit decreased dendritic spine density”

– Result, line 144: “indicating that stabilized Rai14 protects F-actin from destruction in dendritic spines.” – “indicating that Rai14 protects dendritic spines from the pressure of elimination by actin destabilization.”

– Figure 6 legend, line 1147: “The Rai14 cluster at the spine neck contributes to maintaining spines, probably by stabilizing F-actin, thereby upregulating dendritic spine density.” – “The Rai14 cluster at the spine neck contributes to maintaining spines, thereby upregulating dendritic spine density.”

– Result, line 191-192: “Taken together, these results support the importance of Rai14 in the plastic changes of neuronal connections relevant to depressive-like behaviors” – “Taken together, these results support the link between the Rai14-controlled dendritic spine dynamics and depressive-like behaviors.”

– Discussion, line 236-239: “Unlike the spine head filled with branched F-actin and a pool of G-actin at dynamic equilibrium (Hotulainen and Hoogenraad, 2010), the spine neck consists of actin in the form of a linear F-actin bundle and periodic F-actin with a ring structure (Bar et al., 2016; Bucher et al., 2020).” – “Unlike the spine head filled with branched F-actin and a pool of G-actin at dynamic equilibrium (Hotulainen and Hoogenraad, 2010), the spine neck consists of actin in the form of a linear F-actin and periodic F-actin with a ring structure (Bar et al., 2016; Bucher et al., 2020).”

Reviewer #1 (Recommendations for the authors):1. The Figure legends are generally lacking in experimental details, including the brain regions, cell types, and mouse ages analyzed.

In response to the reviewer’s concern, we added information on brain regions, cell types, time points, and spine classification criteria analyzed in figure legends and the Materials and Method section. Please also see the Essential Revisions (for the authors) #1.

2. The authors should provide some explanation of how Rai14 might be regulating gene expression. The gene expression analysis was done on whole brain WT and Rai14+/- mice. Does the brain region matter?

It is possible that decreased spine number and reduced synaptic activity in Rai14 deficient neurons may cause a cellular environment reflecting depression, such as dysregulation of activity-regulated gene expression (Li et al., 2015; Manning et al., 2017; Nestler, 2015; Reul, 2014). In glutamatergic neurons, Ca^2+^ entry in the postsynaptic spine activates NF-κB, one of the activity-regulated transcription factors, and induces its translocation from the activated synapse to the nucleus, where it regulates its target gene expression, including synapse-related genes (Kaltschmidt and Kaltschmidt, 2009; Snow et al., 2014). Besides, Rai14 was identified as one of the factors associated with NF-κB (Li et al., 2014). In TNF‑α stimulated glioblastoma cells, Rai14 is involved in NF-κB activation, translocation to the nucleus, and target gene expression (Shen et al., 2019). Thus, it is possible that Rai14 in the dendritic spine may be involved in the activation of NF-κB in response to excitatory stimulation. Indeed, bioinformatics analysis using JASPAR Predicted Transcription Factor Targets dataset predicted that half of the genes common in both significant DEGs in *Rai14*^+/-^ mouse brains and Aston-Major depressive disorder-DN gene set as NF-κB target genes (ABCA2, ASPA, CNTN2, GPR37, HTR4, MAG, PMP22, RASGRF1, and SORBS2). (Rouillard et al., 2016). Therefore, it is intriguing to speculate that Rai14 may have more direct roles in driving gene expression related to mood control in physiological conditions, bridging the synaptic and nuclear events.

–In response to the reviewer’s suggestion, we added the information related to NF-κB in the text as follows.

– Discussion, line 268-272: “Interestingly, bioinformatics analysis using JASPAR Predicted Transcription Factor Targets dataset predicted that half of those 18 genes, including the aforementioned spine-associated genes (CNTN2, GPR37, HTR4, SORBS2, RASGRF1, ABCA2, ASPA, MAG, PMP22) as target genes of NF-κB (Rouillard et al., 2016), activity-regulated transcription factor (Kaltschmidt and Kaltschmidt, 2009; Snow et al., 2014) regulated by Rai14 (Shen et al., 2019).”

– Regarding brain regions, it is a very important question in that the expression of the Rai14 is not restricted to a specific brain region, thereby indicating that the function of Rai14 is less likely to be brain region-specific. Therefore, investigating Rai14 in other contexts reflecting diverse brain functions involving different brain regions will be another imminent task.

Reviewer #2 (Recommendations for the authors):In my opinion the experiments and results very well support the conclusions and the results are well presented. If I would add here something it could be Rai14 localization in dendritic neck imaged with super-resolution techniques, possibly together with actin. This would be interesting but not necessary. It would (maybe) give ideas of mechanistic details, such as, is Rai14 localizing periodically in spine necks like actin rings.This sentence can be rephrased:Lines 236 -239: "share a common characteristic: actin binding and regulation of actin dynamics. Unlike the spine head filled with branched F-actin and a pool of G-actin at dynamic equilibrium (Hotulainen and Hoogenraad, 2010), the spine neck consists of actin in the form of a linear F-actin bundle and periodic F-actin with a ring structure (Bar et al., 2016; Bucher et al., 2020). "I agree that F-actin is mostly linear or in rings in necks but I would be cautious to say linear F-actin bundle. Thus, I would drop the word "bundle" out from this sentence.

In response to the reviewer’s suggestion, we modified the sentence as follows.

– Discussion, line 236-239: “Unlike the spine head filled with branched F-actin and a pool of G-actin at dynamic equilibrium (Hotulainen and Hoogenraad, 2010), the spine neck consists of actin in the form of a linear F-actin bundle and periodic F-actin with a ring structure (Bar et al., 2016; Bucher et al., 2020).” – “Unlike the spine head filled with branched F-actin and a pool of G-actin at dynamic equilibrium (Hotulainen and Hoogenraad, 2010), the spine neck consists of actin in the form of a linear F-actin and periodic F-actin with a ring structure (Bar et al., 2016; Bucher et al., 2020).”

References

Kaltschmidt, B., and Kaltschmidt, C. (2009). NF-kappaB in the nervous system. Cold Spring Harb Perspect Biol *1*, a001271.

Li, X., Zhao, Y., Tian, B., Jamaluddin, M., Mitra, A., Yang, J., Rowicka, M., Brasier, A.R., and Kudlicki, A. (2014). Modulation of gene expression regulated by the transcription factor NF-kappaB/RelA. J Biol Chem *289*, 11927-11944.

Li, Y., Pehrson, A.L., Waller, J.A., Dale, E., Sanchez, C., and Gulinello, M. (2015). A critical evaluation of the activity-regulated cytoskeleton-associated protein (Arc/Arg3.1)'s putative role in regulating dendritic plasticity, cognitive processes, and mood in animal models of depression. Front Neurosci *9*, 279.

Manning, C.E., Williams, E.S., and Robison, A.J. (2017). Reward Network Immediate Early Gene Expression in Mood Disorders. Front Behav Neurosci *11*, 77.

Nestler, E.J. (2015). Role of the Brain's Reward Circuitry in Depression: Transcriptional Mechanisms. Int Rev Neurobiol *124*, 151-170.

Peng, Y.F., Mandai, K., Sakisaka, T., Okabe, N., Yamamoto, Y., Yokoyama, S., Mizoguchi, A., Shiozaki, H., Monden, M., and Takai, Y. (2000). Ankycorbin: a novel actin cytoskeleton-associated protein. Genes Cells *5*, 1001-1008.

Qian, X., Mruk, D.D., and Cheng, C.Y. (2013a). Rai14 (retinoic acid induced protein 14) is involved in regulating f-actin dynamics at the ectoplasmic specialization in the rat testis. PLoS One *8*, e60656.

Qian, X., Mruk, D.D., Cheng, Y.H., and Cheng, C.Y. (2013b). RAI14 (retinoic acid induced protein 14) is an F-actin regulator: Lesson from the testis. Spermatogenesis *3*, e24824.

Reul, J.M. (2014). Making memories of stressful events: a journey along epigenetic, gene transcription, and signaling pathways. Front Psychiatry *5*, 5.

Rouillard, A.D., Gundersen, G.W., Fernandez, N.F., Wang, Z.C., Monteiro, C.D., McDermott, M.G., and Ma'ayan, A. (2016). The harmonizome: a collection of processed datasets gathered to serve and mine knowledge about genes and proteins. Database-Oxford.

Seipel, K., O'Brien, S.P., Iannotti, E., Medley, Q.G., and Streuli, M. (2001). Tara, a novel F-actin binding protein, associates with the Trio guanine nucleotide exchange factor and regulates actin cytoskeletal organization. J Cell Sci *114*, 389-399.

Shen, X., Zhang, J., Zhang, X., Wang, Y., Hu, Y., and Guo, J. (2019). Retinoic Acid-Induced Protein 14 (RAI14) Promotes mTOR-Mediated Inflammation Under Inflammatory Stress and Chemical Hypoxia in a U87 Glioblastoma Cell Line. Cell Mol Neurobiol *39*, 241-254.

Snow, W.M., Stoesz, B.M., Kelly, D.M., and Albensi, B.C. (2014). Roles for NF-kappaB and gene targets of NF-kappaB in synaptic plasticity, memory, and navigation. Mol Neurobiol *49*, 757-770.

Woo, Y., Kim, S.J., Suh, B.K., Kwak, Y., Jung, H.J., Nhung, T.T.M., Mun, D.J., Hong, J.H., Noh, S.J., Kim, S.*, et al.* (2019). Sequential phosphorylation of NDEL1 by the DYRK2-GSK3beta complex is critical for neuronal morphogenesis. *eLife 8*.